# AlphaPO: Reward Shape Matters for LLM Alignment

**Aman Gupta** [* 1]  **Shao Tang** [* 1]  **Qingquan Song** [1]  **Sirou Zhu** [1]  **Jiwoo Hong** [2]  **Ankan Saha** [1]  **Viral Gupta** [1]  **Noah Lee** [2]  **Eunki Kim** [2]  **Siyu Zhu** [1]  **Parag Agrawal**[† 1]  **Natesh Pillai** [1]  **S. Sathiya Keerthi**[† 1]

## Abstract

Reinforcement Learning with Human Feedback (RLHF) and its variants have made huge strides toward the effective alignment of large language models (LLMs) to follow instructions and reflect human values. More recently, Direct Alignment Algorithms (DAAs) have emerged in which the reward modeling stage of RLHF is skipped by characterizing the reward directly as a function of the policy being learned. Some popular examples of DAAs include Direct Preference Optimization (DPO) and Simple Preference Optimization (SimPO). These methods often suffer from likelihood displacement, a phenomenon by which the probabilities of preferred responses are often reduced undesirably. In this paper, we argue that, for DAAs the reward (function) shape matters. We introduce **AlphaPO**, a new DAA method that leverages an $\alpha$-parameter to help change the shape of the reward function beyond the standard log reward. AlphaPO helps maintain fine-grained control over likelihood displacement and over-optimization. Compared to SimPO, one of the best performing DAAs, AlphaPO leads to about 7% to 10% relative improvement in alignment performance for the instruct versions of Mistral-7B and Llama3-8B while achieving 15% to 50% relative improvement over DPO on the same models. The analysis and results presented highlight the importance of the reward shape and how one can systematically change it to affect training dynamics, as well as improve alignment performance.

*Equal contribution  [1]LinkedIn Corporation, CA, USA [2]KAIST AI, KAIST, South Korea †Work done while at LinkedIn Corporation.  Correspondence to: Aman Gupta <amagupta@linkedin.com>, Shao Tang <shatang@linkedin.com>.

*Proceedings of the $42^{nd}$ International Conference on Machine Learning*, Vancouver, Canada. PMLR 267, 2025. Copyright 2025 by the author(s).

## 1. Introduction

Large language models (LLMs) (Vaswani, 2017; Brown et al., 2020; Dubey et al., 2024) have ushered in a new era for artificial intelligence. LLM training can be broadly split into two stages - pre- and post-training. An important step in the post-training stage is "alignment" - which involves improving the model's ability to follow instructions, human values and style. This step is crucial in helping bridge the gap between the raw ability of pre-trained models and the immense utility of post-trained models.

To this end, researchers have developed several techniques under the umbrella of Reinforcement Learning with Human Feedback (RLHF) (Ouyang et al., 2022; Ziegler et al., 2020). RLHF involves a three stage process - supervised fine-tuning (SFT), reward modeling and reinforcement learning (RL)-based fine-tuning to learn the optimal policy. Learning is typically performed on preference data, which include preferred and dispreferred responses to the same input prompt. While achieving impressive results, RLHF involves several stages and can be cumbersome to train (Casper et al., 2023).

To simplify the alignment process, methods such as Direct Preference Optimization (DPO) (Rafailov et al., 2023) and Simple Preference Optimization (SimPO) (Meng et al., 2024) have been developed. DPO optimizes directly for the optimal policy, bypassing the reward modeling step. The optimal policy is identified as the solution to an expected reward maximization problem while ensuring that the policy does not diverge too much from a reference policy. SimPO achieves the same goal with a different reward function normalized by the length of the generation, and also introduces a margin term to better separate preferred and dispreferred responses. Both belong to a class of methods known as Direct Alignment Algorithms (DAAs) (Rafailov et al., 2024a). DAAs include DPO and its variants (Rafailov et al., 2023; Park et al., 2024a), SimPO (Meng et al., 2024) and methods like CPO (Xu et al., 2024) and ORPO (Hong et al., 2024) among others.

Several studies have shown that DAAs tend to widen the margin between preferred and dispreferred responses while simultaneously reducing the probabilities of preferred responses. Although slightly lower completion likelihood is known to improve output diversity and generalization, ex-

**Generalized Reward Shaping with AlphaPO**

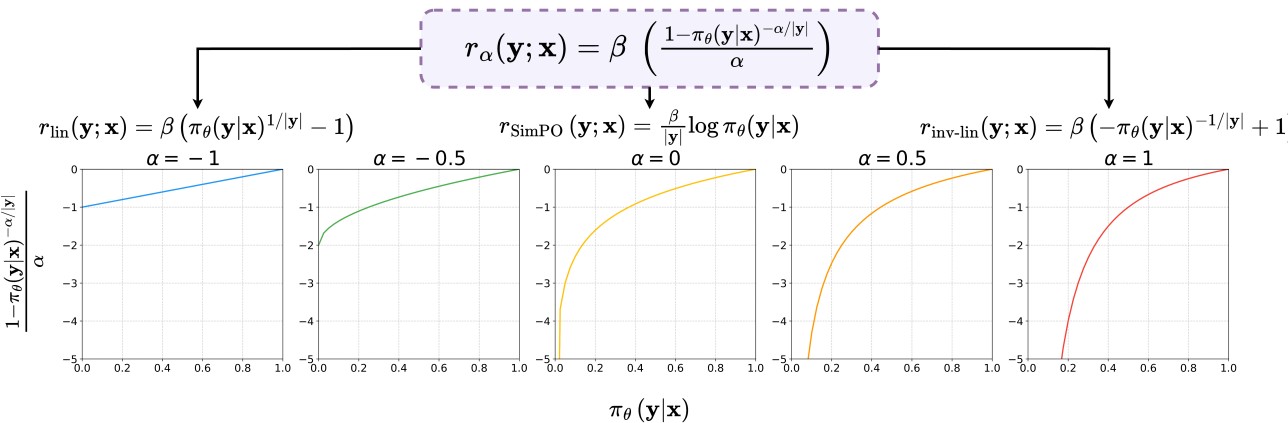

*Figure 1.* The generalized reward shaping paradigm of AlphaPO. By adjusting the parameter $\alpha$ in the reward function $r_\alpha(\mathbf{y}; \mathbf{x})$, AlphaPO induces various reward shapes, leading to different preference optimization dynamics and downstream performances.

cessive focus on increasing the reward margin can degrade actual performance, as measured by human evaluations. This is known as *reward over-optimization* (Shi et al., 2024). Similarly, the reduction in preferred response probabilities, known as *likelihood displacement* (Razin et al., 2024), has been linked to unintended misalignment, further harming performance. Extended training exacerbates both issues, leading to deterioration in the overall effectiveness of preference optimization.

Although the aforementioned issues may be concerning, in practice, there are typically model checkpoints along the training trajectory that demonstrate strong alignment performance. These checkpoints can be identified through early stopping. SimPO, known for its strong alignment capabilities, uses just one epoch of training (Meng et al., 2024). By varying three hyperparameters—learning rate, $\beta$ (reward scaling), and $\gamma$ (reward margin shift)—SimPO generates a three-dimensional manifold of trajectories. These hyperparameters are fine-tuned to enhance alignment performance on benchmarks such as AlpacaEval 2.0 (Dubois et al., 2024), optimizing the win rate to select the final model of choice.

In this paper, we introduce a new method **AlphaPO** based on a fourth important dimension - the reward function shape. Most DAAs, including SimPO, are based on the log reward function. We point out that changing this reward shape yields new types of trajectories with distinctly different margin and preferred-likelihood behaviors. We take inspiration from Wang et al. (2024a), use the reward function arising from the $\alpha$-divergence idea and apply length-normalization to it to generate an interesting class of reward shapes that are parameterized by $\alpha$ as depicted in Figure 1 ($\alpha = 0$ yields the log reward shape for SimPO, making it a special case. More details are provided later.). By choosing an $\alpha$ value different from zero, alignment performance can be significantly im-

proved. We note that although AlphaPO's $\alpha$ reward shapes are inspired by f-DPO(Wang et al., 2024a), they are distinctly different since we use length normalization in the reward and f-DPO does not. Moreover, we demonstrate that non-zero values of $\alpha$ improve generalization performance while f-DPO fails to demonstrate that for any value of $\alpha$ except 0 (which is standard DPO).

Through gradient analysis and experiments, we show how the shape of the newly introduced reward function affects the aggressive or conservative nature of likelihood displacement. For varying values of $\alpha$, we then demonstrate how AlphaPO's performance evolves on evaluation benchmarks such as AlpacaEval 2.0 and ArenaHard (Li et al., 2024). The critical factor is the combination of the $\alpha$-parameterized scoring function with length normalization. We demonstrate that AlphaPO comprehensively outperforms SimPO and DPO across models such as Mistral-7B Instruct (Jiang et al., 2023a) and Llama3-8B Instruct (Dubey et al., 2024), while remaining competitive for models like Gemma2-9B Instruct (Team et al., 2024). In particular, AlphaPO achieves a relative improvement of 7% to 10% over SimPO (and 15% to 50% over DPO) on AlpacaEval 2 for Llama3-8B and Mistral-7B. Additionally, integrating AlphaPO with the SPPO method (Wu et al., 2024) yields strong improvements on AlpacaEval 2.0 with a length controlled win rate of 47.42% for PairRM-based regeneration (Jiang et al., 2023b) of the UltraFeedback dataset (Cui et al., 2024), bettering the results of SimPO with SPPO applied on top.

## 2. Preliminaries

Let us consider the setup of alignment training. $\mathbf{x} = [x_1, x_2, x_3, ....]$ denotes a single input prompt with a sequence of tokens. The objective of an LLM is to generate a relevant and consistent response $\mathbf{y} = [y_1, y_2, y_3, ....]$ to $\mathbf{x}$.

LLM learns a policy $\pi_{\boldsymbol{\theta}}$ parameterized by $\boldsymbol{\theta}$ where $\pi_{\boldsymbol{\theta}}(\mathbf{y}|\mathbf{x})$ denotes the probability assigned to $\mathbf{y}$.

RLHF aims to fine-tune the policy $\pi_{\boldsymbol{\theta}}$ to maximize the reward $r(\cdot)$ without excessively diverging from the initial policy $\pi_{\mathrm{ref}}$ (Christiano et al., 2017; Ouyang et al., 2022).

In the offline alignment learning scenario, the data consist of the prompt $\mathbf{x}$ and a pair of (preferred, dispreferred) responses $(\mathbf{y}_w, \mathbf{y}_l)$[1] of sequences, with preference denoted by $\mathbf{y}_w \succ \mathbf{y}_l$ (Rafailov et al., 2023). This triplet is often denoted as $(\mathbf{x}, \mathbf{y}_w, \mathbf{y}_l)$. RLHF-based methods consist of using supervised fine-tuning (SFT) on a dataset, reward modeling and then reinforcement learning to finally learn an aligned model. Methods such as Direct Preference Optimization (DPO) (Rafailov et al., 2023) , SimPO (Meng et al., 2024), CPO (Xu et al., 2024), and ORPO (Hong et al., 2024) use the Bradley-Terry (BT) (Bradley & Terry, 1952) setup: $p(\mathbf{y}_w \succ \mathbf{y}_l) = \sigma(r(\mathbf{y}_w; \mathbf{x}) - r(\mathbf{y}_l; \mathbf{x}))$, where $\sigma(x) = 1/(1 + \exp(-x))$ is the sigmoid function. These method directly specify the reward function as a function of the optimal policy, as described in the subsequent section. These methods are also known as Direct Alignment Algorithms (DAAs) (Rafailov et al., 2024a). DAAs have achieved immense popularity recently for their simplicity and effectiveness at achieving strong alignment performance (Lambert et al., 2024).

## 2.1. Direct Preference Optimization (DPO)

DPO (Rafailov et al., 2023) is a recent popular method that solves the problem of maximizing the expected reward with an added KL penalty between the sequence probabilities of the model being trained and a reference model $\pi_{\mathrm{ref}}$ (trained using SFT). DPO uses a closed-form expression for the reward $r$ by leveraging the optimality conditions of the problem: $r_{\mathrm{DPO}}(\mathbf{y}; \mathbf{x}) = \beta \log\left(\pi_{\boldsymbol{\theta}}(\mathbf{y}|\mathbf{x}) / \pi_{\mathrm{ref}}(\mathbf{y}|\mathbf{x})\right) + \beta \log(Z(\mathbf{x}))$, where $\pi_{\boldsymbol{\theta}}$ refers to the policy being trained and $Z(\mathbf{x})$ is the partition function. Plugging this into the BT model, we get the following loss function for DPO:

$$L_{\mathrm{DPO}} = -\mathbb{E}_{(\mathbf{x}, \mathbf{y}_w, \mathbf{y}_l) \sim \mathcal{D}}\left[ \log\sigma\left( \beta\log\frac{\pi_{\boldsymbol{\theta}}(\mathbf{y}_w|\mathbf{x})}{\pi_{\mathrm{ref}}(\mathbf{y}_w|\mathbf{x})} \right.\right.$$
$$\left.\left. - \beta\log\frac{\pi_{\boldsymbol{\theta}}(\mathbf{y}_l|\mathbf{x})}{\pi_{\mathrm{ref}}(\mathbf{y}_l|\mathbf{x})} \right) \right], \quad (1)$$

where $\mathcal{D}$ denotes the preference dataset.

## 2.2. Simple Preference Optimization (SimPO)

SimPO (Meng et al., 2024) is a modification of DAAs like DPO. SimPO scales the reward by the length of the output[2]:

---

[1]Some papers refer to this as the *(chosen, reject) pair*.

[2]The idea of using average log probability for reward was originally proposed in Hong et al. (2024, ORPO).

$r_{\mathrm{SimPO}}(\mathbf{y}; \mathbf{x}) = \frac{\beta}{|\mathbf{y}|} \log \pi_{\boldsymbol{\theta}}(\mathbf{y}|\mathbf{x})$, where $|\mathbf{y}|$ is the length of $\mathbf{y}$. SimPO also introduces an additional margin term in the BT objective: $p(\mathbf{y}_w \succ \mathbf{y}_l) = \sigma(r(\mathbf{y}_w; \mathbf{x}) - r(\mathbf{y}_l; \mathbf{x}) - \gamma)$. This results in the following loss for SimPO:

$$L_{\mathrm{SimPO}} = -\mathbb{E}_{(\mathbf{x}, \mathbf{y}_w, \mathbf{y}_l) \sim \mathcal{D}}\left[ \log\sigma\left( \frac{\beta}{|\mathbf{y}_w|} \log\pi_{\boldsymbol{\theta}}(\mathbf{y}_w|\mathbf{x}) \right.\right.$$
$$\left.\left. - \frac{\beta}{|\mathbf{y}_l|} \log\pi_{\boldsymbol{\theta}}(\mathbf{y}_l|\mathbf{x}) - \gamma \right) \right]. \quad (2)$$

While the parameter $\gamma$ influences the size of margin values, the parameter $\beta$ determines how close $-\log\sigma(\beta z)/\beta$ is to the hinge loss, i.e., $\max\{-z, 0\}$. SimPO demonstrates its effectiveness as a robust and efficient alignment method for LLMs due to several key innovations. Unlike DPO, SimPO eliminates the reliance on a reference model during training, reducing memory and computational overhead while ensuring reward maximization aligns with inference. Furthermore, SimPO establishes length normalization, first introduced by ORPO, as a critical factor for alignment performance. This technique enables SimPO to outperform DPO and other methods on benchmarks like AlpacaEval2.0 (Dubois et al., 2024), achieving superior results without increasing response length. Notably, SimPO's length normalization has inspired enhancements in other methods, such as Tulu (Lambert et al., 2024), underscoring its impact.

By fixing training epochs to one and carefully tuning $\beta$ and $\gamma$, SimPO achieves strong alignment performance. A key aspect of its robustness lies in its constant penalization $\gamma$, which addresses limitations in DPO. While DPO parameterizes the optimal human preference distribution under the BT model with a maximum likelihood objective, it relies on divergence penalties from a reference policy $\pi_{\mathrm{ref}}$, which can be suboptimal when preference samples are sampled from arbitrary policies (Tunstall et al., 2024) or $\pi_{\mathrm{ref}}$ itself (Meng et al., 2024; Lambert et al., 2024). SimPO mitigates this issue by replacing the variable divergence penalty $-\beta\log\left(\pi_{\mathrm{ref}}(\mathbf{y}_w|\mathbf{x})/\pi_{\mathrm{ref}}(\mathbf{y}_l|\mathbf{x})\right)$ in $L_{\mathrm{DPO}}$ with a constant $\gamma$, thereby improving robustness under noisy preference samples.

## 3. Reward Function Matters

In this section, we dive deep into the shape of reward functions and its relationship with likelihood displacement. The main premise of this paper is that by systematically modifying the $\log$ reward function, it is possible to adjust the intensity of the likelihood displacement and, in particular, improve the alignment performance of SimPO.

### 3.1. Likelihood displacement

Let us define the *length normalized margin* as $(1/|\mathbf{y}_w|) \log \pi_w - (1/|\mathbf{y}_l|) \log \pi_l$, where $\pi_w \stackrel{\triangle}{=} \pi_{\boldsymbol{\theta}}(\mathbf{y}_w|\mathbf{x})$, $\pi_l \stackrel{\triangle}{=} \pi_{\boldsymbol{\theta}}(\mathbf{y}_w|\mathbf{x})$. Clearly, minimization of the preference loss in (2) would (in expectation) lead to an increase in the length normalized margin. Although this seems to imply that training decreases $\log \pi_l$ and increases $\log \pi_w$, usually $\log \pi_l$ rapidly decreases while also dragging the preferred likelihood $\log \pi_w$ down. This phenomenon is called *likelihood displacement* (Razin et al., 2024). Although several recent papers (Pal et al., 2024; Liu et al., 2024; Pang et al., 2024; Yuan et al., 2024; Rafailov et al., 2024b; Tajwar et al., 2024) mention this behavior and discuss its ramifications, (Razin et al., 2024) dissect this in detail. They make a key observation - since both $\log \pi_w$ and $\log \pi_l$ decrease, other responses (denoted by $z$) increase in likelihood. If $z$ is as preferable as $\mathbf{y}_w$ (e.g., $z$ is similar in meaning to $\mathbf{y}_w$), then the likelihood displacement is *benign*. However, if $z$ is a poor response, then likelihood displacement is labeled as *catastrophic* and leads to unintentional consequences.

A closely related idea is discussed in (Shi et al., 2024), where the authors note that high likelihood of preferred responses and big gaps between the likelihood of preferred and dispreferred responses can lead to *reward over-optimization* - resulting in a better margin but worse generalization. A controlled lowering of completion likelihood is shown to be beneficial for output diversity and generalization.

The ideas listed above indicate the importance of balancing the optimization of the reward margin with the likelihood of the preferred and dispreferred responses. To avoid confusion, we use the phrase "likelihood displacement" to refer to all the related ideas described above.

While excessive likelihood displacement (over-optimization) is detrimental, controlled likelihood displacement offers clear benefits, as demonstrated by methods like SimPO, which employ early stopping and limit training to one or a few epochs. Building on these insights, we explore how DAAs can be designed to shape the margin and preferred/dispreferred likelihoods.

### 3.2. The AlphaPO reward function

Most DAAs are based on the $\log$ reward function. We begin by discussing f-DPO (Wang et al., 2024a) where the authors point out that standard DPO corresponds to using reverse KL as the divergence and that other divergences can lead to improvement in properties like generation diversity. One parametric divergence proposed by them is the $\alpha$-divergence. This divergence has the reverse Kullback–Leibler (KL) divergence as a limiting case when $\alpha \to 0$, and the forward KL divergence as a limiting case when $\alpha \to 1$ (Cichocki

et al., 2008). However, in the DPO context, the paper does not find much benefit from this extension since standard DPO ($\alpha = 0$) yields the best alignment performance.

f-DPO operates by allowing alignment to incorporate various divergences, which determines the implicit base reward function. We consider a similar idea in the context of SimPO - we use the reward function corresponding to $\alpha$-divergence, but crucially modify it with length-normalization:

$$r_\alpha(\mathbf{y}; \mathbf{x}) = \beta \left( \frac{1 - \pi_{\boldsymbol{\theta}}(\mathbf{y}|\mathbf{x})^{-\alpha/|\mathbf{y}|}}{\alpha} \right). \tag{3}$$

This yields the following preference loss in the context of the margin-based BT model:

$$L_{\text{AlphaPO}} = - \mathbb{E}_{(\mathbf{x}, \mathbf{y}_w, \mathbf{y}_l) \sim \mathcal{D}} \left[ \log \sigma \left( \frac{-\beta}{\alpha} \pi_{\boldsymbol{\theta}}(\mathbf{y}_w|\mathbf{x})^{-\frac{\alpha}{|\mathbf{y}_w|}} \right. \right.$$
$$\left. \left. + \frac{\beta}{\alpha} \pi_{\boldsymbol{\theta}}(\mathbf{y}_l|\mathbf{x})^{-\frac{\alpha}{|\mathbf{y}_l|}} - \gamma \right) \right]. \tag{4}$$

We refer to this new method, derived from utilizing the new reward function, as **AlphaPO**. In the limit $\alpha \to 0$, the AlphaPO reward function yields SimPO's $\log$ reward. Specific choices $\alpha = 1$ and $\alpha = -1$ yield the inverse-linear and linear reward functions, respectively: $r_{\text{inv-lin}}(\mathbf{y}; \mathbf{x}) = \beta \left( -1/\overline{\pi}_{\boldsymbol{\theta}}(\mathbf{y}|\mathbf{x}) + 1 \right)$ and $r_{\text{lin}}(\mathbf{y}; \mathbf{x}) = \beta \left( \overline{\pi}_{\boldsymbol{\theta}}(\mathbf{y}|\mathbf{x}) - 1 \right)$, where $\overline{\pi}_{\boldsymbol{\theta}}(\mathbf{y}|\mathbf{x}) = \pi_{\boldsymbol{\theta}}(\mathbf{y}|\mathbf{x})^{1/|\mathbf{y}|}$.

AlphaPO differs from f-DPO in the following ways:

**(1) Length normalization -** AlphaPO uses length normalization - a crucial element in improving SimPO's performance over DPO.

**(2) Fewer restrictions on the value of $\alpha$ -** f-DPO requires the estimation of the normalization constant $Z(x)$ for some divergences. Specifically for $\alpha$-divergence, f-DPO restricts the value of $\alpha \in (0, 1)$ for the normalization constant to cancel out. No such restrictions are required for AlphaPO.

**(3) Improved generalization with tuning of $\alpha$ -** While f-DPO fails to demonstrate that any value of $\alpha$ except 0 (which is standard DPO) yields improved generalization, we demonstrate in the experiments section that non-zero values of $\alpha$ actually improve generalization performance.

It turns out that a simple condition on $\alpha$ yields reward function shapes that encourage likelihood displacement. For details, please see Lemma A.1 and Figure 4 in Appendix A.5.

### 3.3. Understanding AlphaPO using gradient analysis and experiments

We now perform gradient analysis and conduct experiments to study the three key *length-normalized* quantities: preferred likelihood $(1/|\mathbf{y}_w|) \log \pi_w$, dispreferred likelihood

$(1/|\mathbf{y}_l|) \log \pi_l$ and margin $(1/|\mathbf{y}_w|) \log \pi_w - (1/|\mathbf{y}_l|) \log \pi_l$. This will help us understand the degree of likelihood displacement that happens across methods, the role of $\alpha$ and the effect on alignment performance.

Let us begin with a gradient analysis of AlphaPO for a single training example. For $\mathbf{y}, \mathbf{x}$ such that $\pi_{\boldsymbol{\theta}}(\mathbf{y}|\mathbf{x}) > 0$, we have

$$r'_\alpha(\pi_{\boldsymbol{\theta}}(\mathbf{y}|\mathbf{x})) \triangleq \frac{\partial r_\alpha(\mathbf{y};\mathbf{x})}{\partial \pi_{\boldsymbol{\theta}}(\mathbf{y}|\mathbf{x})} = \frac{\beta \exp\left(-\alpha \frac{\log \pi_{\boldsymbol{\theta}}(\mathbf{y}|\mathbf{x})}{|\mathbf{y}|}\right)}{\pi_{\boldsymbol{\theta}}(\mathbf{y}|\mathbf{x})|\mathbf{y}|} \tag{5}$$

where the last expression on the right hand side has been written with the purpose of clearly separating out the role of $\alpha$. Let $v$ be a generic weight parameter. Define non-negative parameters $c_w = -\frac{\log \pi_w}{|\mathbf{y}_w|}, c_l = -\frac{\log \pi_l}{|\mathbf{y}_l|}$. Define $\delta r = r_\alpha(\pi_w) - r_\alpha(\pi_l)$. Based on (3) and (5), the magnitude of the gradient of per-sample loss $\ell$ with respect to $v$ is:

$$\left|\frac{\partial \ell}{\partial v}\right| = |\ell'(\delta r - \gamma)| \cdot \left|r'_w \frac{\partial \pi_w}{\partial v} - r'_l \frac{\partial \pi_l}{\partial v}\right|, \tag{6}$$

where the reward derivatives for the dispreferred and preferred responses are: $r'_l \triangleq r'_\alpha(\pi_l) = \beta \exp(\alpha c_l)/(\pi_l |\mathbf{y}_l|)$ and $r'_w \triangleq r'_\alpha(\pi_w) = \beta \exp(\alpha c_w)/(\pi_w |\mathbf{y}_w|)$ and $\delta r = \frac{\beta}{\alpha}[\exp(\alpha c_l) - \exp(\alpha c_w)]$. Substituting these quantities into Equation (6), we obtain $|\partial \ell/\partial v| = T_1(\alpha) \cdot T_2(\alpha)$ where

$$T_1(\alpha) = \frac{\beta}{1 + \exp\left(\frac{\beta}{\alpha}\left[\exp(\alpha c_l) - \exp(\alpha c_w)\right] - \gamma\right)}, \tag{7}$$

$$T_2(\alpha) = \left|\frac{\exp(\alpha c_w)}{\pi_w |\mathbf{y}_w|} \frac{\partial \pi_w}{\partial v} - \frac{\exp(\alpha c_l)}{\pi_l |\mathbf{y}_l|} \frac{\partial \pi_l}{\partial v}\right|. \tag{8}$$

We are interested in studying how $\alpha$ affects $|\partial \ell/\partial v|$. It turns out that $T_1(\alpha)$ and $T_2(\alpha)$ exhibit markedly different behaviors as functions of $\alpha$. This discrepancy leads to $|\partial \ell/\partial v|$ behaving in a non-monotonic fashion with respect to $\alpha$. The limiting behavior is established in Theorem 3.1, whose proof is deferred to Appendix A.1.

**Theorem 3.1.** *Let $\beta > 0$ and $\gamma \geq 0$ be fixed constants. Let* $\Delta = \frac{1}{\pi_w |\mathbf{y}_w|} \frac{\partial \pi_w}{\partial v} - \frac{1}{\pi_l |\mathbf{y}_l|} \frac{\partial \pi_l}{\partial v}$. *Then:*

1. $\displaystyle\lim_{\alpha \to -\infty} \left|\frac{\partial \ell}{\partial v}\right| = 0.$

2. *As $\alpha \to +\infty$, the limit of $\left|\frac{\partial \ell}{\partial v}\right|$ depends on the sign of the margin:*

   – *If the margin is positive, then $\displaystyle\lim_{\alpha \to +\infty} \left|\frac{\partial \ell}{\partial v}\right| = 0.$*

   – *If the margin is non-positive, $\displaystyle\lim_{\alpha \to +\infty} \left|\frac{\partial \ell}{\partial v}\right| = +\infty$*
   *except when the margin is zero and $\Delta = 0$ [3].*

---

[3]If the margin is zero and $\Delta = 0$, $T_2$ is identically zero, i.e., $T_2(\alpha) \equiv 0$ and thus $|\partial \ell/\partial v| = 0$.

As a consequence, large $|\alpha|$ values impose a regularization effect on alignment training due to the vanishing gradient for samples with positive length-normalized margins. The regularization effects are stronger for positive $\alpha$ values (super-exponential) compared to negative $\alpha$ values (exponential); see the proof for details. The following numerical illustration adequately demonstrates the limiting behavior in Theorem 3.1.

**Illustration 1 - Positive margin.** Consider a toy example with $\beta = 1$, $\gamma = 0$, $|\mathbf{y}_w| = |\mathbf{y}_l| = 1$, $\partial \pi_w/\partial v = \partial \pi_l/\partial v = 1$, $\log \pi_w = -1$, and $\log \pi_l = -2$. Substituting these values, we find $|\partial \ell/\partial v| = (1 + \exp((e^{2\alpha} - e^\alpha)/\alpha))^{-1} |e^{\alpha+1} - e^{2\alpha+2}|$. For the values of $\alpha = -2$, 0, 0.25, 1, and 2, the corresponding magnitudes of the derivative $|\partial \ell/\partial v|$ are 0.11, 1.26, 1.63, 0.44, and $2.15 \times 10^{-8}$, respectively. The associated values for $T_1$ are 0.49, 0.27, 0.19, 0.01, and $5.60 \times 10^{-11}$, while the $T_2$ values are 0.23, 4.67, 8.69, 47.21, and 383.34.

**Illustration 2 - Negative margin.** Consider the toy example, where $\log \pi_w = -2$ and $\log \pi_l = -1$, the magnitude of the derivative is given by: $|\partial \ell/\partial v| = \left(1 + \exp\left((e^\alpha - e^{2\alpha})/\alpha\right)\right)^{-1} |e^{2+2\alpha} - e^{1+\alpha}|$. Using the same $\alpha$ values, the computed $|\partial \ell/\partial v|$ values are 0.12, 3.41, 7.05, 46.77, and 383.34, respectively. The corresponding values for $T_1$ are 0.51, 0.73, 0.81, 0.99, and 1.00, while the $T_2$ values remain identical to those in Illustration 1.

These results demonstrate non-monotonic behavior in the relationship between $|\partial \ell/\partial v|$ and $\alpha$ (see Appendix A.10 for a holistic illustration).

The key takeaways from the analysis are: (a) reward functions significantly impact preference-based alignment learning, and (b) changes in $\alpha$ notably influence the training process by altering gradients and modifying the relative importance assigned to preferred and dispreferred responses. To gain a deeper understanding, we examine the evolution of $\pi_{\boldsymbol{\theta}(t)}(\mathbf{y}_w|\mathbf{x})$ during training under gradient flow. Specifically, we formalize how this relative importance influences the dynamics of $\pi_{\boldsymbol{\theta}(t)}(\mathbf{y}_w|\mathbf{x})$ over training time $t \geq 0$, where $\boldsymbol{\theta}(t)$ denotes the model parameters at time $t$. For brevity, we define $\pi_w \triangleq \pi_{\boldsymbol{\theta}(t)}(\mathbf{y}_w|\mathbf{x})$, $\pi_l \triangleq \pi_{\boldsymbol{\theta}(t)}(\mathbf{y}_l|\mathbf{x})$ and $\nabla \pi_w \triangleq \nabla_{\boldsymbol{\theta}(t)} \pi_w, \nabla \pi_l \triangleq \nabla_{\boldsymbol{\theta}(t)} \pi_l$.

**Theorem 3.2** (Evolution of $\pi_w$). *Consider a single training example $(\mathbf{x}, \mathbf{y}_w, \mathbf{y}_l)$, where $\mathbf{y}_w$ and $\mathbf{y}_l$ are the preferred and dispreferred responses, respectively. Consider $\pi_w, \pi_l \in (0, 1)$. Assume that $\|\nabla \pi_w\| \neq 0$. If*

$$\frac{r'(\pi_w)}{r'(\pi_l)} \geq \frac{\langle \nabla \pi_w, \nabla \pi_l \rangle}{\|\nabla \pi_w\|^2}, \tag{9}$$

*then under gradient-flow dynamics, the probability of the preferred response $\pi_w$ increases over time, i.e. $\frac{d}{dt} \pi_w \geq 0$.*

The proof is in Appendix A.2.

If $\langle \nabla \pi_w, \nabla \pi_l \rangle \leq 0$, then (9) holds trivially for all $\alpha$, and $\pi_w$ increases. A more interesting case is where $\langle \nabla \pi_w, \nabla \pi_l \rangle > 0$, i.e., the gradient step will increase or decrease the probabilities for both $\pi_w$ and $\pi_l$. In this senario, we have the following Corollary (proof in Appendix A.3).

**Corollary 3.3.** *Under the conditions of Theorem 3.2, suppose that $\langle \nabla \pi_w, \nabla \pi_l \rangle > 0$. If the margin is negative, then for $\alpha \geq \alpha_0$, the probability $\pi_w$ increases over time. Conversely, if the margin is positive, then for $\alpha \leq \alpha_0$, the probability $\pi_w$ increases over time, where $\alpha_0$ is defined as*

$$\alpha_0 = -\frac{\log \left( \frac{\|\nabla \pi_w\|^2}{\langle \nabla \pi_w, \nabla \pi_l \rangle} \cdot \frac{\pi_w |\mathbf{y}_w|}{\pi_l |\mathbf{y}_l|} \right)}{\frac{1}{|\mathbf{y}_w|} \log \pi_w - \frac{1}{|\mathbf{y}_l|} \log \pi_l}. \quad (10)$$

As pointed out by (Fisch et al., 2024), methods such as DPO have global optima that include policies capable of shifting nearly all probability mass to responses that never appear in the training set—and even assigning near-zero probability to all training data responses that correspond to winning generations. In Theorem 3.2 and Corollary 3.3, we demonstrate that such likelihood displacement can be mitigated by ensuring a sufficiently large ratio of $r'(\pi_w)/r'(\pi_l)$, controlled by selecting appropriate values of $\alpha$.

The above analysis is at the level of a single example. A detailed analysis of the interaction of many examples is hard. Therefore, we conduct training experiments using the instruct versions of the Mistral-7B and Gemma-2-9B models on the UltraFeedback dataset with 1 epoch of training (detailed setup in Section 4). Specifically, we vary $\alpha$ from $-2.0$ to $2.0$, while keeping other hyperparameters fixed.

Line plots for the length-normalized quantities of the Mistral model can be found in Figure 2 (a similar figure for Gemma 2 (Figure 7) and the corresponding box plots Figures 15 and 16 are in Appendix A.11). As shown in the figure, the influence of $\alpha$ on dynamics is indeed non-monotonic. Specifically, there are larger displacements for $\alpha$ values close to $0$ and smaller displacements for larger $|\alpha|$ values. In particular, larger $|\alpha|$s lead to smaller quartile values of margin toward the end of training, which could help mitigate the over-optimization. Scatter plots of preferred likelihood vs. margin in Figures 13 and 14 tell a similar story. For large absolute values of $\alpha$ ($-2$ and $2$), we notice a clear regularization effect for both margin and preferred likelihood.

During the initial phase of training, many samples exhibit an undesired ordering, characterized by a negative margin. According to Corollary 3.3, selecting a sufficiently large $\alpha$ should lead to an increase in $\pi_w$. This behavior is evident in epochs 0 to 0.1 in Figures 2 and 7 (for $\alpha > -2$).

Additionally, we explore in Appendix A.4 what happens when a reference model is included in the SimPO and AlphaPO training objectives.

## 4. Experiments

### 4.1. Experimental setup

**Models** We conduct all our experiments using the instruct versions of three popular families of models - Llama 3(Dubey et al., 2024), Mistral (Jiang et al., 2023a) and Gemma 2 (Team et al., 2024). We make this choice for two reasons - (1) The aforementioned models represent the state-of-the-art and (2) they have been used in several recent works, making comparisons with baselines easier.

**Datasets** We chose the UltraFeedback (UF) dataset (Cui et al., 2024) for all experiments. Previous works (Meng et al., 2024; Wu et al., 2024; Zhao et al., 2024) have demonstrated that using an *on-policy* setting for the instruct setup helps mitigate the distribution shift between off-the-shelf instruct variants of these models and the preference optimization process. Following (Meng et al., 2024), specifically, we regenerate five responses for every prompt in the UF dataset using a sampling temperature of 0.8. We then use two reward models - PairRM (Jiang et al., 2023b) and ArmoRM (Wang et al., 2024b) to rank the 5 responses. The highest scoring response is labeled $\mathbf{y}_w$ and the lowest scoring response is labeled $\mathbf{y}_l$. We use the PairRM-based dataset to conduct experiments for Llama 3 and Mistral, and leverage the ArmoRM-based dataset for Llama 3 and Gemma 2-based experiments.

**Hyperparameter tuning** AlphaPO introduces a new parameter $\alpha$ as described in equation (3). We tune $\alpha$ for each different LLM and note the effect of tuning alpha in section 4.2. The exact values of learning rate, $\alpha$, $\beta$, $\gamma$ and other hyperparameters used in the experiment iterations are detailed in Appendix A.6.

**Baselines** We compare AlphaPO primarily against SimPO and DPO since they represent the state-of-the-art.

**Evaluation.** We evaluate trained models using two popular benchmarks - AlpacaEval 2.0 (Dubois et al., 2024) and Arena-Hard (Wang et al., 2024b) (referred to as AE2 and AH, respectively, hereinafter). AE2 consists of 805 prompts sourced from a variety of datasets, gearing it towards measuring the ability of models to follow diverse and complex instructions. For AE2, we report WR (win rate) and LC (length-controlled win rate). LC is specifically designed to discourage models from using verbose answers, since GPT4 is known to favor longer responses when judging for instruction following (Wang et al., 2023; Park et al., 2024b). For AH, we only report WR since it doesn't offer LC.

### 4.2. Results

**AlphaPO outperforms SimPO and DPO on most benchmarks** as demonstrated in Table 1. Focusing on PairRM-based results, AlphaPO outperforms SimPO and DPO across

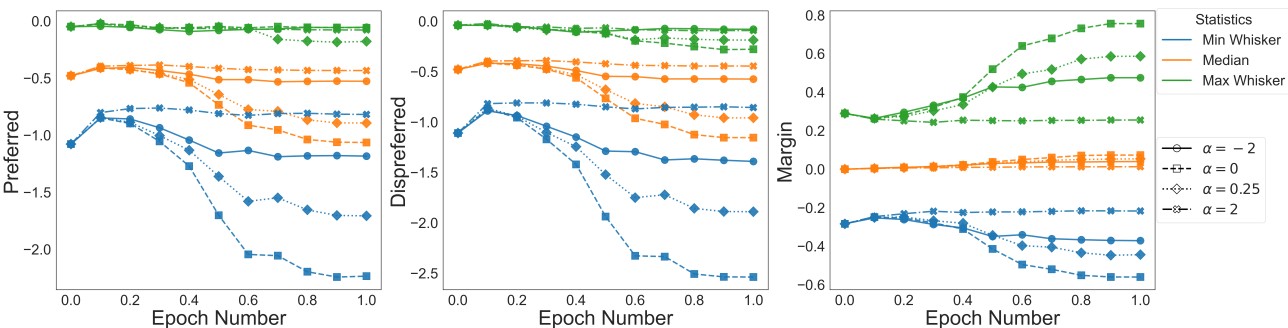

*Figure 2.* Tracking various statistics for length-normalized preferred likelihood, dispreferred likelihood and margin over an epoch for the **Mistral** instruct model. The min and max are computed after removing outliers.

*Table 1.* Comparison of DPO, SimPO, and AlphaPO for on-policy data created using PairRM and ArmoRM.

| | PairRM | | | | | | ArmoRM | | | | | |
|---|---|---|---|---|---|---|---|---|---|---|---|---|
| | Llama-3-8B-Instruct | | | Mistral-7B-Instruct | | | Llama-3-8B-Instruct | | | Gemma-2-9B-Instruct | | |
| | AE2 | | AH | AE2 | | AH | AE2 | | AH | AE2 | | AH |
| **Method** | LC (%) | WR (%) | WR (%) | LC (%) | WR (%) | WR (%) | LC (%) | WR (%) | WR (%) | LC (%) | WR (%) | WR (%) |
| **DPO** | 39.18 | 36.84 | 34.1 | 21.96 | 22.16 | 14.8 | 47.19 | 45.61 | 35.5 | 67.83 | 65.77 | **60.7** |
| **SimPO** | 42.05 | 36.90 | 32.8 | 29.71 | 31.12 | 21.5 | 51.66 | 46.54 | 35.4 | 73.72 | **67.36** | 59.0 |
| **AlphaPO** | **45.37** | **40.97** | **34.6** | **33.03** | **34.12** | **21.7** | **53.01** | **47.64** | **36.0** | **74.13** | 65.89 | 57.7 |

both AE2 and AH for the instruct versions of the Llama 3 and Mistral models. The results are especially pronounced for AE2 where AlphaPO relatively improves over SimPO by 7% to 10% for LC, and over DPO by 15% for Llama 3 and 50% for Mistral. Notably, AlphaPO doesn't increase the generation length significantly for all the models when compared to SimPO and DPO (more in Appendix A.7).

For ArmoRM-based results, Llama 3 AlphaPO outperforms both DPO and SimPO. For Gemma 2, we note that AlphaPO has a better LC and a slightly lower or comparable WR for AE2 when compared to SimPO and DPO. The lower WR for AE2 is also mirrored by AH results where SimPO and DPO are slightly better. This is consistent with the observation that GPT as a judge has a bias towards longer responses, and AH doesn't have a way to control for length when measuring rewards (Wang et al., 2023; Park et al., 2024b). More detailed results can be found in Appendix A.7. From Figure 5 in Appendix A.8, we demonstrate that the number of AE2 samples where AlphaPO outperforms SimPO is higher than the number of samples where SimPO does better.

**Effect of $\alpha$ on generalization performance** To understand the effect of $\alpha$ on model generalization, we measure the AE2 performance of various models with changing $\alpha$. Results for the Mistral model are in Figure 3 (results for other models are similar and are in Figure 8 in Appendix A.11). A slightly positive value of $\alpha$ achieves the best AE2 performance, with a drop-off on either side of

*Table 2.* Language modeling benchmark evaluation results for SimPO and AlphaPO. We evaluated in 10-shot setting for HellaSwag.

| | Llama-3-8B-Instruct | | Mistral-7B-Instruct | |
|---|---|---|---|---|
| | SimPO | AlphaPO | SimPO | AlphaPO |
| **HellaSwag** | 0.7576 | **0.7694** | 0.8610 | **0.8638** |
| **TruthfulQA** | 0.6078 | **0.6142** | 0.7061 | **0.7127** |

the peak value. The drop-off is less steep on the positive side. This is not surprising, since a positive $\alpha$ results in less aggressive likelihood displacement when compared to SimPO (which lowers both preferred and dispreferred likelihoods) and subsequently achieves higher AE2 performance. A negative $\alpha$ results in more aggressive likelihood placement compared to positive values of $\alpha$. See section 3.3 for more discussion.

**Effect of tuning $\gamma$ on generalization performance** We examine the impact of the margin parameter $\gamma$ on AE2 performance. To this end, we tune $\gamma$ while keeping other hyperparameters fixed at the optimal settings reported in Table 3. We use PairRM-based data for all models, except for Gemma 2, which employs ArmoRM-based data.

Results for the Mistral model appear in Figure 3 (findings for other models are in Figure 9 in Appendix A.11). Consistent with the findings of (Meng et al., 2024), increasing $\gamma$

improves AE2 performance to a certain level, beyond which larger values hurt performance. This indicates that there exists an optimal gamma that has to be tuned depending on the model family and the quality of response generation (determined by AE2 evaluation) is not determined by the margin alone. On the other hand, increasing $\gamma$ leads to longer response lengths, as it dominates the length-normalized quantities $r(\mathbf{y}_w; \mathbf{x}) - r(\mathbf{y}_l; \mathbf{x})$ in the loss function.

**Effect on reward scores** To better understand the qualitative impact of various alignment methods, we look at the reward distribution for the test set of UltraFeedback using PairRM. We compare AlphaPO to SimPO and SFT for the Mistral in Figure 3. It is evident that AlphaPO yields a reward distribution that is better or on par with SimPO (the strongest baseline in our experiments). Since the intent of alignment methods is to match human preferences, this test is a good proxy for measuring the performance (see the right plot in Figure 12 in Appendix A.11 for the Llama 3 result).

**Evolution of KL Divergence with training** In Figures 10 and 11 of the Appendix A.11, we present the evolution of KL divergence (relative to the SFT checkpoint) and the AE2 LC during training for AlphaPO and SimPO applied to both Mistral and Llama 3 instruct models. We observe that both methods remain close to the instruct model, exhibiting minimal divergence even without explicit regularization to the reference policy. Notably, AlphaPO achieves a KL divergence comparable to that of SimPO while attaining better LC values, suggesting that excessively high or low KL divergence can impede generalization. Furthermore, although KL divergence continues to increase and eventually plateaus, a larger KL does not necessarily lead to improved LC. These findings highlight the importance of early stopping and careful tuning of the total number of training steps to enhance preference generalization performance.

**Combining SimPO/AlphaPO and SPPO** SimPO and AlphaPO use BT modeling to diverge from a reference model, achieving strong alignment through optimal hyperparameters. In contrast, SPPO (Wu et al., 2024) cautiously moves toward a Nash equilibrium. For the PairRM-based (Jiang et al., 2023b) experiments, we select the best performing checkpoints for SimPO and AlphaPO and train both further using SPPO (see Appendix A.9 for details). Without extensive tuning, SPPO improves the AE2 LC of both SimPO (from 42.05 to 45.06) and AlphaPO (from 45.37 to 47.42). This demonstrates that methods like SPPO can be orthogonally used to improve these methods.

**Additional experiments on other datasets** To understand the impact of AlphaPO on datasets other than those used traditionally for alignment, we compare AlphaPO and SimPO-based checkpoints of the Mistral and Llama

models on HellaSwag (Zellers et al., 2019) and TruthfulQA (Lin et al., 2021). HellaSwag is a multiple-choice commonsense reasoning benchmark whereas TruthfulQA is a question-answering benchmark designed to evaluate a model's propensity to generate truthful versus misleading or false answers. Results are presented in Table 2. AlphaPO outperforms SimPO across all settings.

## 5. Related work

**Reinforcement learning with human feedback** RLHF aligns language models to human preferences by leveraging parameterized reward models as proxies and reinforcement learning (RL) algorithms (Christiano et al., 2017; Ziegler et al., 2019). Having parameterized reward models trained with the Bradley-Terry model (Bradley & Terry, 1952), algorithms like proximal policy optimization (Schulman et al., 2017) and REINFORCE (Williams, 1992) are applied for language models to maximize the reward with online generations (Ziegler et al., 2020; Ouyang et al., 2022; Ahmadian et al., 2024; Kazemnejad et al., 2024). While typically being used to align the style of generations to human preferences (Ziegler et al., 2020; Ouyang et al., 2022), the paradigm of RLHF is being applied to specialized tasks like complex mathematical reasoning (Kazemnejad et al., 2024; Lambert et al., 2024) and coding (Gehring et al., 2024).

**Direct alignment algorithms** Direct alignment algorithms (Rafailov et al., 2024a, DAAs) as methods for directly aligning the language models to human preferences bypass RL in RLHF (Yuan et al., 2023; Zhao et al., 2023; Rafailov et al., 2023; Wang et al., 2024a; Xu et al., 2024; Gheshlaghi Azar et al., 2024; Ethayarajh et al., 2024; Wu et al., 2024; Hong et al., 2024; Meng et al., 2024) either through optimizing expected rewards penalized by divergence from a reference policy (*e.g.,* DPO (Rafailov et al., 2023), SPPO (Wu et al., 2024)) or directly as a function of the policy (*e.g.,* ORPO (Hong et al., 2024), SimPO (Meng et al., 2024)).

While these score functions effectively guide fine-tuning with alignment data, they often exhibit over-optimization, excessively widening the margin between chosen and rejected outputs, sometimes lowering the quality of chosen responses. Recent research like f-divergence Preference Optimization (Han et al., 2024, f-PO) offers theoretical insights addressing over-optimization by minimizing f-divergences.

## 6. Conclusion

Altering the reward function shape to enhance the alignment performance of SimPO is a novel insight introduced and demonstrated in this paper. Since reward function (via the choice of divergence measure) occurs in other methods such

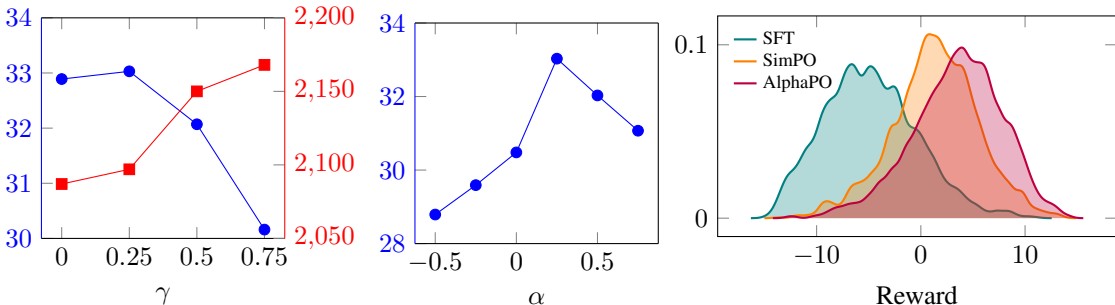

*Figure 3.* Ablation studies on the Mistral instruct model. **Left**: Effect of the margin parameter $\gamma$ on the AE2 LC (blue) and response length (red). **Middle**: Effect of the parameter $\alpha$ on the AE2 LC. **Right**: Reward distributions for the SFT, SimPO, and AlphaPO checkpoints.

as DPO and RLHF, whether it is possible that changing its shape can provide value for them too is a worthwhile future research direction.

## Impact Statement

This paper presents work whose goal is to advance the field of Machine Learning. There are many potential societal consequences of our work, none which we feel must be specifically highlighted here.

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

# A. Appendix

## A.1. Proof of Theorem 3.1

*Proof.* For $\pi_w, \pi_l \in (0,1)$, recall $c_w = -\frac{\log \pi_w}{|\mathbf{y}_w|} > 0, c_l = -\frac{\log \pi_l}{|\mathbf{y}_l|} > 0$.

**Part 1:** $\alpha \to -\infty$.

Since $c_l > 0$ and $c_w > 0$, both $\exp(\alpha\, c_l)$ and $\exp(\alpha\, c_w)$ go to 0. Hence

$$\exp(\alpha\, c_l) \;-\; \exp(\alpha\, c_w) \;\to\; 0, \quad \tfrac{\beta}{\alpha}\left[\exp(\alpha\, c_l) - \exp(\alpha\, c_w)\right] \;\to\; 0.$$

Therefore

$$T_1(\alpha) \;=\; \frac{\beta}{1 + \exp\!\left(\frac{\beta}{\alpha}[\exp(\alpha\, c_l) - \exp(\alpha\, c_w)] - \gamma\right)} \;\to\; \frac{\beta}{1 + e^{-\gamma}},$$

a finite and positive constant.

For $T_2(\alpha)$:

$$T_2(\alpha) = \left|\exp(\alpha\, c_w)\, \frac{1}{\pi_w |\mathbf{y}_w|}\frac{\partial \pi_w}{\partial v} \;-\; \exp(\alpha\, c_l)\, \frac{1}{\pi_l |\mathbf{y}_l|}\frac{\partial \pi_l}{\partial v}\right|.$$

Each term contains a factor $\exp(\alpha\, c_i)$ which goes to 0, so $T_2(\alpha) \to 0$. Thus the product satisfies

$$\lim_{\alpha \to -\infty}\left|\frac{\partial \ell}{\partial v}\right| \;=\; \lim_{\alpha \to -\infty} T_1(\alpha)\, T_2(\alpha) \;=\; 0.$$

**Part 2:** $\alpha \to +\infty$.

**Case 2(a), Positive margin:** $c_l > c_w$. Since $c_l > c_w$, $\exp(\alpha c_l) \gg \exp(\alpha c_w)$ as $\alpha \to +\infty$. Thus:

$$\frac{\beta}{\alpha}\left[\exp(\alpha c_l) - \exp(\alpha c_w)\right] \sim \frac{\beta}{\alpha}\exp(\alpha c_l). \tag{11}$$

The denominator of $T_1(\alpha)$ satisfies:

$$\exp\left(\frac{\beta}{\alpha}\left[\exp(\alpha c_l) - \exp(\alpha c_w)\right] - \gamma\right) \sim \exp\left(\frac{\beta}{\alpha}\exp(\alpha c_l)\right). \tag{12}$$

Since $\exp(\alpha c_l) \gg \exp(\alpha c_w)$, the term $\exp(\alpha c_l)$ dominates:

$$T_2(\alpha) \sim \frac{\exp(\alpha c_l)}{\pi_l |\mathbf{y}_l|}\left|\frac{\partial \pi_l}{\partial v}\right|. \tag{13}$$

Combining the asymptotics of $T_1(\alpha)$ and $T_2(\alpha)$, we find:

$$T_1(\alpha)T_2(\alpha) \sim \frac{\beta}{\exp\left(\frac{\beta}{\alpha}\exp(\alpha c_l)\right)} \times \frac{\exp(\alpha c_l)}{\pi_l |\mathbf{y}_l|}\left|\frac{\partial \pi_l}{\partial v}\right| \tag{14}$$

$$\sim \frac{\beta \exp(\alpha c_l)}{\exp\left(\frac{\beta}{\alpha}\exp(\alpha c_l)\right)\pi_l |\mathbf{y}_l|}\left|\frac{\partial \pi_l}{\partial v}\right| \to 0. \tag{15}$$

Thus, we conclude:

$$\lim_{\alpha \to +\infty}\left|\frac{\partial \ell}{\partial v}\right| = 0, \quad \text{when } c_l > c_w. \tag{16}$$

**Case 2(b), Negative Margin:** $c_l < c_w$. Then $\exp(\alpha\, c_w)$ dominates $\exp(\alpha\, c_l)$, and

$$\exp(\alpha\, c_l) - \exp(\alpha\, c_w) \;\approx\; -\exp(\alpha\, c_w).$$

Hence

$$\tfrac{\beta}{\alpha}\left[\exp(\alpha\,c_l) - \exp(\alpha\,c_w)\right] \;\approx\; -\tfrac{\beta}{\alpha}\,\exp(\alpha\,c_w) \;\to\; -\infty,$$

so

$$\exp\!\left(-\tfrac{\beta}{\alpha}\exp(\alpha\,c_w) - \gamma\right) \;\to\; 0, \quad T_1(\alpha) \;\to\; \beta.$$

In $T_2(\alpha)$, the $\exp(\alpha\,c_w)$ term dominates, so $T_2(\alpha)$ grows on the order of $\exp(\alpha\,c_w)$. Thus

$$T_2(\alpha) \;\to\; +\infty, \quad T_1(\alpha)\,T_2(\alpha) \;\sim\; \beta\,\exp(\alpha\,c_w) \;\to\; +\infty,$$

implying $\lim_{\alpha\to+\infty}\left|\frac{\partial\ell}{\partial v}\right| = +\infty$ when $c_l < c_w$.

**Case 2(c), Zero Margin:** $c_l = c_w$. Then

$$\exp(\alpha\,c_l) - \exp(\alpha\,c_w) \;=\; 0, \quad T_1(\alpha) \;=\; \frac{\beta}{1 + \exp(-\gamma)},$$

which is a positive constant. In $T_2(\alpha)$,

$$T_2(\alpha) = \left|\exp(\alpha\,c)\,\frac{1}{\pi_w\,|\mathbf{y}_w|}\frac{\partial\pi_w}{\partial v} - \exp(\alpha\,c)\,\frac{1}{\pi_l\,|\mathbf{y}_l|}\frac{\partial\pi_l}{\partial v}\right|$$

$$= \exp(\alpha\,c)\left|\frac{1}{\pi_w\,|\mathbf{y}_w|}\frac{\partial\pi_w}{\partial v} - \frac{1}{\pi_l\,|\mathbf{y}_l|}\frac{\partial\pi_l}{\partial v}\right|.$$

If $\Delta = \frac{1}{\pi_w\,|\mathbf{y}_w|}\frac{\partial\pi_w}{\partial v} - \frac{1}{\pi_l\,|\mathbf{y}_l|}\frac{\partial\pi_l}{\partial v} \neq 0$, then $T_2(\alpha)$ grows exponentially since $\exp(\alpha\,c) \to +\infty$. Hence the product with the finite positive $T_1(\alpha)$ diverges to $+\infty$.

Thus

$$\lim_{\alpha\to+\infty}\left|\frac{\partial\ell}{\partial v}\right| = +\infty \quad \text{when } \Delta \neq 0.$$

$\square$

## A.2. Proof of Theorem 3.2

*Proof.* As the model parameters evolve according to gradient flow:

$$\frac{d}{dt}\boldsymbol{\theta}(t) = -\nabla\ell(\boldsymbol{\theta}(t)), \tag{17}$$

where $\ell(\boldsymbol{\theta})$ is a differentiable loss function. By the chain rule, the time derivative of $\pi_w$ is

$$\frac{d}{dt}\pi_w = \langle\nabla\pi_w, \frac{d}{dt}\boldsymbol{\theta}(t)\rangle = -\langle\nabla\pi_w, \nabla\ell(\boldsymbol{\theta}(t))\rangle. \tag{18}$$

Recall the loss gradient

$$\nabla\ell(\boldsymbol{\theta}(t)) = \ell'(\delta r - \gamma)\left[r'(\pi_w)\nabla\pi_w - r'(\pi_l)\nabla\pi_l\right], \tag{19}$$

where $\ell'(\delta r - \gamma)$ is negative. Applying the condition Substituting into the expression for $\frac{d}{dt}\pi_w$ gives

$$\frac{d}{dt}\pi_w = -\ell'(\delta r - \gamma)\left[r'(\pi_w)\|\nabla\pi_w\|^2 - r'(\pi_l)\langle\nabla\pi_w, \nabla\pi_l\rangle\right]. \tag{20}$$

Factoring out $r'(\pi_l)$, we obtain

$$\frac{d}{dt}\pi_w = -\ell'(\delta r - \gamma)r'(\pi_l)\left[\frac{r'(\pi_w)}{r'(\pi_l)}\|\nabla\pi_w\|^2 - \langle\nabla\pi_w, \nabla\pi_l\rangle\right]. \tag{21}$$

Recall (5), $r'(\pi_l) > 0$. Since $-\ell'(\delta r - \gamma) > 0$, the sign of $\frac{d}{dt}\pi_w$ is determined by

$$\frac{r'(\pi_w)}{r'(\pi_l)}\|\nabla\pi_w\|^2 - \langle\nabla\pi_w, \nabla\pi_l\rangle. \tag{22}$$

Rearranging the inequality condition, we require

$$\frac{r'(\pi_w)}{r'(\pi_l)} \geq \frac{\langle \nabla \pi_w, \nabla \pi_l \rangle}{\|\nabla \pi_w\|^2}, \tag{23}$$

to ensure that the bracketed term is nonnegative. Under this condition, it follows that

$$\frac{d}{dt}\pi_w \geq 0. \tag{24}$$

Thus, the probability $\pi_w$ of the preferred response increases over time. $\qquad\square$

### A.3. Proof of Corollary 3.3

*Proof.* Under the assumption that $\langle \nabla \pi_w, \nabla \pi_l \rangle > 0$, the condition in Theorem 3.2 can be satisfied by appropriately choosing the ratio $\alpha$. The ratio is parameterized by $\alpha$

$$\frac{r'(\pi_w)}{r'(\pi_l)} = \frac{\pi_l |\mathbf{y}_l|}{\pi_w |\mathbf{y}_w|} \exp\left(-\alpha\left(\frac{\log \pi_w}{|\mathbf{y}_w|} - \frac{\log \pi_l}{|\mathbf{y}_l|}\right)\right). \tag{25}$$

Substituting (25) into the inequality (9), we obtain

$$\frac{\pi_l |\mathbf{y}_l|}{\pi_w |\mathbf{y}_w|} \exp\left(-\alpha\left(\frac{\log \pi_w}{|\mathbf{y}_w|} - \frac{\log \pi_l}{|\mathbf{y}_l|}\right)\right) \geq \frac{\langle \nabla \pi_w, \nabla \pi_l \rangle}{\|\nabla \pi_w\|^2}. \tag{26}$$

Rearranging terms, we have

$$\exp\left(-\alpha\left(\frac{\log \pi_w}{|\mathbf{y}_w|} - \frac{\log \pi_l}{|\mathbf{y}_l|}\right)\right) \geq \frac{\|\nabla \pi_w\|^2}{\langle \nabla \pi_w, \nabla \pi_l \rangle} \cdot \frac{\pi_w |\mathbf{y}_w|}{\pi_l |\mathbf{y}_l|}. \tag{27}$$

Taking the natural logarithm of both sides yields

$$-\alpha\left(\frac{\log \pi_w}{|\mathbf{y}_w|} - \frac{\log \pi_l}{|\mathbf{y}_l|}\right) \geq \log\left(\frac{\|\nabla \pi_w\|^2}{\langle \nabla \pi_w, \nabla \pi_l \rangle} \cdot \frac{\pi_w |\mathbf{y}_w|}{\pi_l |\mathbf{y}_l|}\right). \tag{28}$$

Solving for $\alpha$,

$$\alpha \leq \alpha_0 \quad \text{if} \quad \frac{\log \pi_w}{|\mathbf{y}_w|} > \frac{\log \pi_l}{|\mathbf{y}_l|} \quad \text{(positive margin)}, \tag{29}$$

and

$$\alpha \geq \alpha_0 \quad \text{if} \quad \frac{\log \pi_w}{|\mathbf{y}_w|} < \frac{\log \pi_l}{|\mathbf{y}_l|} \quad \text{(negative margin)}, \tag{30}$$

where $\alpha_0$ is defined as

$$\alpha_0 = -\frac{\log\left(\frac{\|\nabla \pi_w\|^2}{\langle \nabla \pi_w, \nabla \pi_l \rangle} \cdot \frac{\pi_w |\mathbf{y}_w|}{\pi_l |\mathbf{y}_l|}\right)}{\frac{\log \pi_w}{|\mathbf{y}_w|} - \frac{\log \pi_l}{|\mathbf{y}_l|}}. \tag{31}$$

$$\square$$

### A.4. AlphaPO with Reference Policy

Another interesting observation is what happens when a reference model is included in the SimPO and AlphaPO training objectives. Interestingly, the SimPO variant reduces to SimPO with a per-response-pair margin, whereas the AlphaPO variant reduces to AlphaPO with per-response weights for the preferred and dispreferred responses.

The inclusion of the reference policy in SimPO (2) results in the loss

$$L_{\text{SimPO w ref}} = -\mathbb{E}_{(\mathbf{x}, \mathbf{y}_w, \mathbf{y}_l) \sim \mathcal{D}} \left[\log \sigma\left(\frac{\beta}{|\mathbf{y}_w|}\log\left(\frac{\pi_\theta(\mathbf{y}_w|\mathbf{x})}{\pi_{\text{ref}}(\mathbf{y}_w|\mathbf{x})}\right) - \frac{\beta}{|\mathbf{y}_l|}\left(\log\frac{\pi_\theta(\mathbf{y}_l|\mathbf{x})}{\pi_{\text{ref}}(\mathbf{y}_l|\mathbf{x})}\right) - \gamma\right)\right]$$

$$= -\mathbb{E}_{(\mathbf{x}, \mathbf{y}_w, \mathbf{y}_l) \sim \mathcal{D}} \left[\log \sigma\left(\frac{\beta}{|\mathbf{y}_w|}\log \pi_\theta(\mathbf{y}_w|\mathbf{x}) - \frac{\beta}{|\mathbf{y}_l|}\log \pi_\theta(\mathbf{y}_l|\mathbf{x}) - \gamma'(\mathbf{y}_w, \mathbf{y}_l, \mathbf{x})\right)\right].$$

Therefore, $L_{\text{SimPO w ref}}$ reduces to SimPO with the per-response-pair margin $\gamma'(\mathbf{y}_w, \mathbf{y}_l, \mathbf{x})$

$$\gamma'(\mathbf{y}_w, \mathbf{y}_l, \mathbf{x}) = \gamma + \frac{\beta}{|\mathbf{y}_w|} \log \pi_{\text{ref}}(\mathbf{y}_w|\mathbf{x}) - \frac{\beta}{|\mathbf{y}_l|} \log \pi_{\text{ref}}(\mathbf{y}_l|\mathbf{x}).$$

In contrast, with the reference policy, the AlphaPO (4) becomes

$$L_{\text{AlphaPO w ref}} = -\mathbb{E}_{(\mathbf{x}, \mathbf{y}_w, \mathbf{y}_l) \sim \mathcal{D}} \left[ \log \sigma \left( \frac{-\beta}{\alpha} \left( \frac{\pi_{\boldsymbol{\theta}}(\mathbf{y}_w|\mathbf{x})}{\pi_{\text{ref}}(\mathbf{y}_w|\mathbf{x})} \right)^{-\alpha/|\mathbf{y}_w|} + \frac{\beta}{\alpha} \left( \frac{\pi_{\boldsymbol{\theta}}(\mathbf{y}_l|\mathbf{x})}{\pi_{\text{ref}}(\mathbf{y}_l|\mathbf{x})} \right)^{-\alpha/|\mathbf{y}_l|} - \gamma \right) \right]$$

$$= -\mathbb{E}_{(\mathbf{x}, \mathbf{y}_w, \mathbf{y}_l) \sim \mathcal{D}} \left[ \log \sigma \left( \frac{-\beta'(\mathbf{y}_w, \mathbf{x})}{\alpha} \pi_{\boldsymbol{\theta}}(\mathbf{y}_w|\mathbf{x})^{-\alpha/|\mathbf{y}_w|} + \frac{\beta'(\mathbf{y}_l, \mathbf{x})}{\alpha} \pi_{\boldsymbol{\theta}}(\mathbf{y}_l|\mathbf{x})^{-\alpha/|\mathbf{y}_l|} - \gamma \right) \right],$$

where

$$\beta'(\mathbf{y}_w, \mathbf{x}) = \beta \, \pi_{\text{ref}}(\mathbf{y}_w|\mathbf{x})^{\alpha/|\mathbf{y}_w|},$$
$$\beta'(\mathbf{y}_l, \mathbf{x}) = \beta \, \pi_{\text{ref}}(\mathbf{y}_l|\mathbf{x})^{\alpha/|\mathbf{y}_l|}.$$

Therefore, $L_{\text{AlphaPO w ref}}$ reduces to AlphaPO with per-response weight $\beta'(\mathbf{y}_w, \mathbf{x})$ and $\beta'(\mathbf{y}_l, \mathbf{x})$.

### A.5. Likelihood displacement with AlphaPO

It turns out that a simple condition on $\alpha$ yields reward function shapes that encourage likelihood displacement (and subsequent decrease in the likelihood of both, preferred and dispreferred responses). The following lemma describes this.

**Lemma A.1** (Monotonically decreasing derivative of $r_\alpha(\pi)$). *Let $\pi \in (0, 1]$ be the probability of a response $\mathbf{y}$ for parameters $\boldsymbol{\theta}$, and let $\alpha, \beta \in \mathbb{R}$ with $\beta > 0$. Also, let $|\mathbf{y}| \geq 1$. Then $r'_\alpha(\pi)$ is monotonically decreasing in $\pi$ if and only if $\alpha \geq -|\mathbf{y}|$.*

We relegate the proof to Appendix A. In Lemma A.1, we use a slight abuse of notation where we refer to the reward function as $r_\alpha(\pi_\theta(\mathbf{y}|\mathbf{x}))$ instead of $r_\alpha(\mathbf{y}|\mathbf{x})$. Since in general, $|\mathbf{y}| > 1$ (and potentially much larger than 1) for responses, most practical values of $\alpha$ will satisfy $\alpha \geq -|\mathbf{y}|$. Thus, by Lemma A.1, lower likelihood values will have higher derivative values for the reward function. This encourages preference optimization to move the likelihood of preferred and dispreferred responses to lower values.

Figure 4 illustrates this for the log reward function ($\alpha = 0$). Notice that for the same difference in likelihood, one can achieve a much higher reward difference if the likelihood of preferred and dispreferred responses are both reduced.

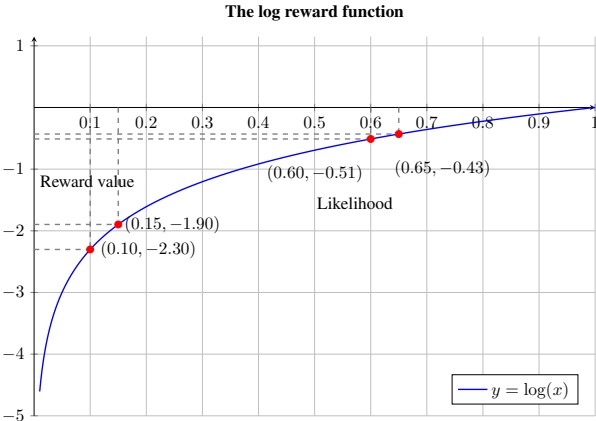

*Figure 4.* The log reward function ($\mathbf{y}$) as a function of the completion likelihood ($\mathbf{x}$). From the plot, it is evident that for a similar change in likelihood, the difference in reward is much higher when the likelihood changes from 0.1 to 0.15 ($\sim 0.40$), rather than from 0.6 to 0.65 ($\sim 0.08$).

#### A.5.1. PROOF OF LEMMA A.1

*Proof.* The derivative of the reward function with respect to $\pi$ is given by

$$r'_\alpha(\pi) = \frac{\beta}{|\mathbf{y}|} \cdot \pi^{-\left(\frac{\alpha}{|\mathbf{y}|}+1\right)}. \tag{32}$$

To determine when $r'_\alpha(\pi)$ is monotonically decreasing, we compute its derivative with respect to $\pi$:

$$\frac{dr'_\alpha(\pi)}{d\pi} = \frac{d}{d\pi}\left(\frac{\beta}{|\mathbf{y}|} \cdot \pi^{-\left(\frac{\alpha}{|\mathbf{y}|}+1\right)}\right) = -\frac{\beta}{|\mathbf{y}|}\left(\frac{\alpha}{|\mathbf{y}|}+1\right)\pi^{-\left(\frac{\alpha}{|\mathbf{y}|}+2\right)}. \tag{33}$$

Since $\beta > 0$, $|\mathbf{y}| \geq 1$, and $\pi^{-\left(\frac{\alpha}{|\mathbf{y}|}+2\right)} > 0$ for $\pi \in (0, 1]$, the sign of $\frac{dr'_\alpha(\pi)}{d\pi}$ is determined by the term $-\left(\frac{\alpha}{|\mathbf{y}|}+1\right)$.

For $r'_\alpha(\pi)$ to be monotonically decreasing in $\pi$, we require:

$$\frac{dr'_\alpha(\pi)}{d\pi} \leq 0.$$

This inequality holds if and only if

$$\frac{\alpha}{|\mathbf{y}|}+1 \geq 0 \quad \Longleftrightarrow \quad \alpha \geq -|\mathbf{y}|.$$

$\square$

## A.6. Implementation Details

**Training hyper-parameter tuning**  Following the recommendations of SimPO, we adopt a global batch size of 128, a maximum sequence length of 2048, and a cosine learning rate schedule with a warmup ratio of 0.1 for one epoch across all training settings. For DPO, we use the best $\beta$ and learning rate values reported in SimPO github.

Although AlphaPO introduces an additional hyperparameter, $\alpha$, compared to SimPO, the need for extensive grid searches can be mitigated if the optimal parameters for SimPO have already been determined. Instead of performing a full grid search, we can leverage the best parameters identified for SimPO and conduct a coordinate-wise hyperparameter search in a greedy manner. This approach allows us to achieve improved performance rapidly, often within a few search iterations. Notably, the optimal parameters of $\beta$, $\gamma$ and learning rate for AlphaPO show only minor deviations from those of SimPO. The hyperparameters that differ from the SimPO settings are highlighted in bold in Table 3. Follow the practice from the SimPO repo, we report $\gamma/\beta$ (instead of $\gamma$) in Table 3 .

*Table 3.* Best hyperparameters for training.

| Method | Model | $\alpha$ | $\beta$ | $\gamma/\beta$ | Learning Rate |
|---|---|---|---|---|---|
| **DPO** | Mistral-Instruct | - | 0.01 | - | $5 \times 10^{-7}$ |
| | Llama-3-Instruct | - | 0.01 | - | $7 \times 10^{-7}$ |
| | Llama-3-Instruct (ArmoRM) | - | 0.01 | - | $3 \times 10^{-7}$ |
| | Gemma-2-Instruct | - | 0.01 | - | $5 \times 10^{-7}$ |
| **SimPO** | Mistral-Instruct | - | 2.5 | 0.1 | $5 \times 10^{-7}$ |
| | Llama-3-Instruct | - | 2.5 | 0.55 | $1 \times 10^{-6}$ |
| | Llama-3-Instruct (ArmoRM) | - | 10.0 | 0.3 | $1 \times 10^{-6}$ |
| | Gemma-2-Instruct | - | 10 | 0.5 | $8 \times 10^{-7}$ |
| **AlphaPO** | Mistral-Instruct | 0.25 | 2.5 | 0.1 | $\mathbf{7 \times 10^{-7}}$ |
| | Llama-3-Instruct | 0.25 | 2.5 | **1.0** | $1 \times 10^{-6}$ |
| | Llama-3-Instruct (ArmoRM) | 0.25 | 10.0 | 0.3 | $\mathbf{1.1 \times 10^{-6}}$ |
| | Gemma-2-Instruct | 0.1 | 10 | 0.5 | $8 \times 10^{-7}$ |

**Decoding hyperparameters**  For AlpacaEval 2.0, we adopt the default settings for AlpacaEval 2.0 with `weighted_alpaca_eval_gpt4_turbo` as the annotator and use `gpt4_turbo` as the reference model. We use a sampling decoding strategy to generate responses, with a temperature of 0.5 for the Mistral-Instruct setting and a temperature of 0.9 for Llama-3-Instruct settings following from the SimPO configs. We use a tempreture of 0.7 following from the WPO-HB config for better reproducibility. For Arena-Hard, we use the default greedy decoding for all settings and methods.

**Computation**  All the training experiments in this paper were conducted on 8×A100 GPUs with the `adamw_torch` optimizer based on the alignment-handbook. The training time for Mistral-Instruct and Llama-3-Instruct is around 2.3 hours, while Gemma-2-Instruct requires 3 hours.

**Open Sourced Models used in Experiments**  The list of open-sourced LLMs used in our experiments and their Huggingface IDs are listed in Table 4.

| Model | Huggingface ID |
|---|---|
| Mistral-Instruct SFT | mistralai/Mistral-7B-Instruct-v0.2 |
| Llama-3-Instruct SFT | meta-llama/Meta-Llama-3-8B-Instruct |
| Gemma-2-Instruct SFT | google/gemma-2-9b-it |

*Table 4.* List of open-source models used in experiments.

## A.7. Detailed results for AlphaPO and SimPO

The full results of the best runs are reported in Table 5.

*Table 5.* Detailed results of AlpacaEval 2 and Arena-Hard for the best run. LC means length-controlled win rate, WR means raw win rate, and STD means standard deviation of win rate over different instructions. Length is the average generation length. For Arena-Hard, we report the win rate and 95% confidence interval.

| Models | AlpacaEval 2 | | | | Arena-Hard | | | |
|---|---|---|---|---|---|---|---|---|
| | LC (%) | WR (%) | STD (%) | Length | WR | 95 CI high | 95 CI low | Length |
| **Mistral-7B-Instruct** | | | | | | | | |
| **DPO** | 21.96 | 22.16 | 1.25 | 2034 | 14.8 | 16.4 | 13.1 | 667 |
| **SimPO** | 29.71 | 31.12 | 1.37 | 2330 | 21.5 | 23.6 | 19.8 | 551 |
| **AlphaPO** | 33.03 | 34.12 | 1.40 | 2097 | 21.7 | 24.0 | 19.9 | 503 |
| **Llama-3-8B-Instruct** | | | | | | | | |
| **DPO** | 39.18 | 36.84 | 1.40 | 1885 | 34.1 | 35.9 | 31.5 | 539 |
| **SimPO** | 42.05 | 36.90 | 1.42 | 1759 | 32.8 | 34.9 | 30.5 | 485 |
| **AlphaPO** | 45.37 | 40.97 | 1.45 | 1820 | 34.6 | 37.1 | 32.9 | 508 |
| **Llama-3-8B-Instruct (ArmoRM)** | | | | | | | | |
| **DPO** | 47.19 | 45.61 | 1.47 | 1974 | 33.4 | 38.0 | 35.5 | 587 |
| **SimPO** | 51.66 | 46.54 | 1.47 | 1806 | 35.4 | 37.8 | 33.3 | 515 |
| **AlphaPO** | 53.01 | 47.64 | 1.47 | 1794 | 36.0 | 38.7 | 33.9 | 507 |
| **Gemma 2-9B-Instruct** | | | | | | | | |
| **DPO** | 67.83 | 65.77 | 1.39 | 2042 | 60.7 | 63.3 | 58.3 | 729 |
| **SimPO** | 73.72 | 67.36 | 1.37 | 1832 | 59.0 | 61.5 | 56.7 | 720 |
| **AlphaPO** | 74.13 | 65.89 | 1.41 | 1802 | 57.7 | 60.4 | 55.6 | 689 |

## A.8. Win Rate Heatmap

In Figure 5, we create AE2 win rate heatmaps of AlphaPO and SimPO for the Mistral instruct and Llama 3 instruct models.

**Observations on Win Rate:** It is observed that the percentage of instances where AlphaPO outperforms the base GPT model, but SimPO does not, is significantly higher compared to the percentage of instances where SimPO outperforms the base GPT model, but AlphaPO does not.

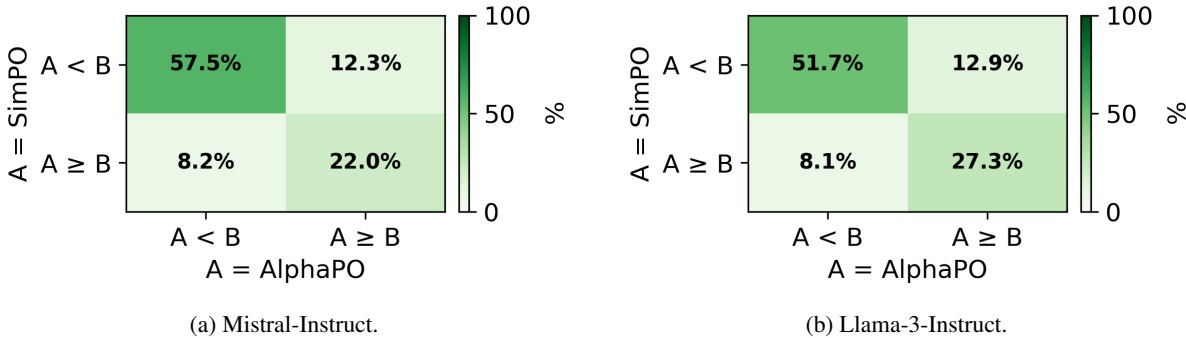

(a) Mistral-Instruct.          (b) Llama-3-Instruct.

*Figure 5.* Win rate heatmap of Mistral-Instruct and Llama-3-Instruct on AlpacaEval 2. B represents the reference model (i.e., `gpt4_turbo`).

### A.9. Training Details of SPPO

We use a 20k sampled subset from the UltraFeedback dataset (Cui et al., 2024) and follow the same hyper parameter setting in the SPPO except for changing the learning rate and number of epochs. We set the learning rate to be $10^{-7}$ with linear warm up and decay and beta to $0.001$ for both methods. We use 6 epochs for SPPO and 4 epochs for AlphaPO.

### A.10. Illustration of the Non-Monotonicity of $|\partial \ell / \partial v|$

To illustrate the non-monotonic behavior of the gradient magnitude $|\partial \ell / \partial v|$, we examine a simple example with

$$\beta = 5, \quad \gamma = 0, \quad |\mathbf{y}_w| = |\mathbf{y}_l|, \quad \frac{\partial \pi_w}{\partial v} = \frac{\partial \pi_l}{\partial v} = 1, \quad \log \pi_w = -5, \quad \log \pi_l = -10.$$

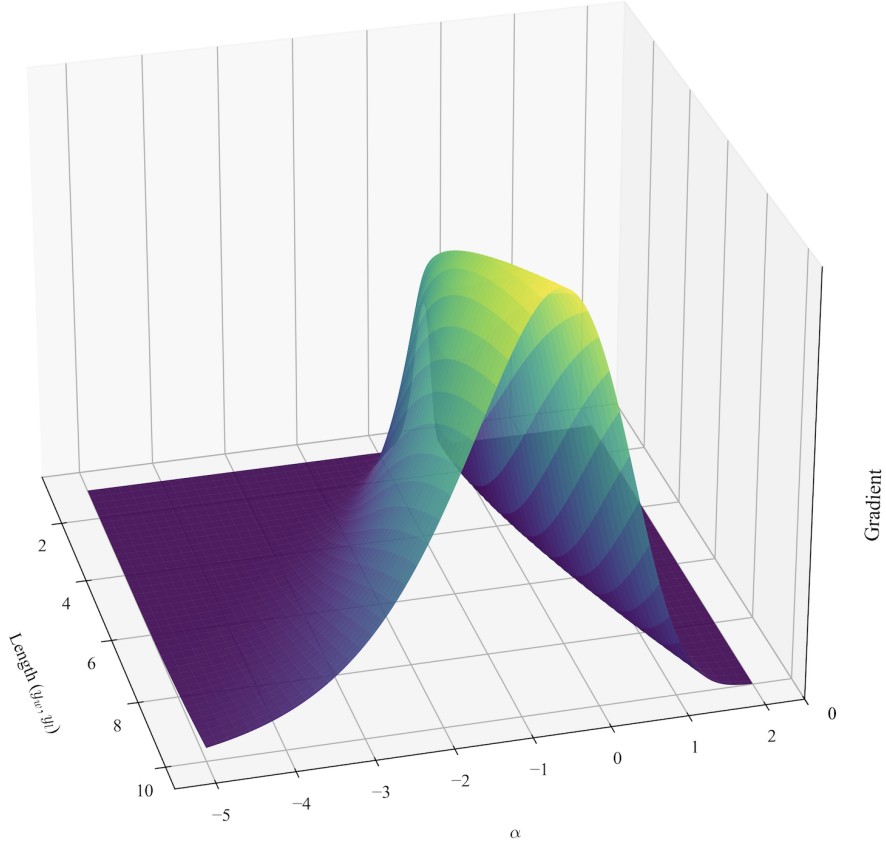

*Figure 6.* Surface plot of the gradient magnitude $|\partial \ell / \partial v|$ ($z$-axis, log scale) as a function of the parameter $\alpha$ ($x$-axis) and the common norm $|\mathbf{y}| = |\mathbf{y}_w| = |\mathbf{y}_l|$ ($y$-axis).

### A.11. Extra plots

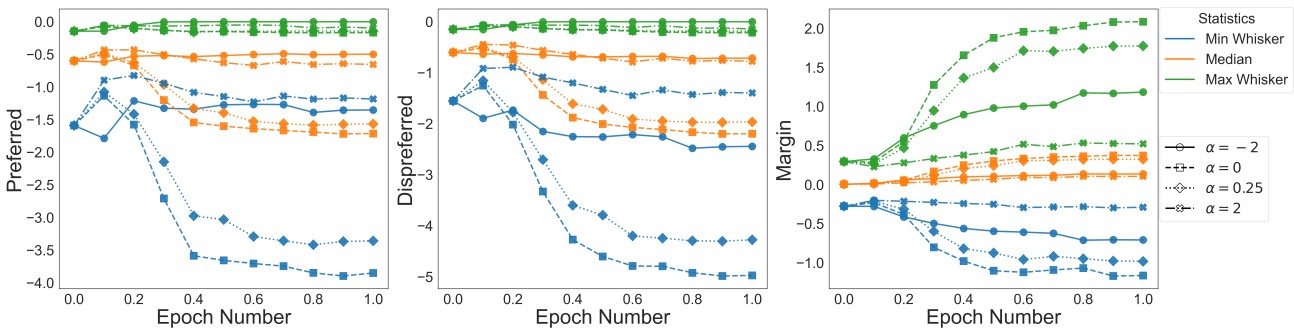

*Figure 7.* Tracking various statistics for length-normalized preferred likelihood, dispreferred likelihood and margin over an epoch for the **Gemma 2** instruct model. The min and max are computed after removing outliers.

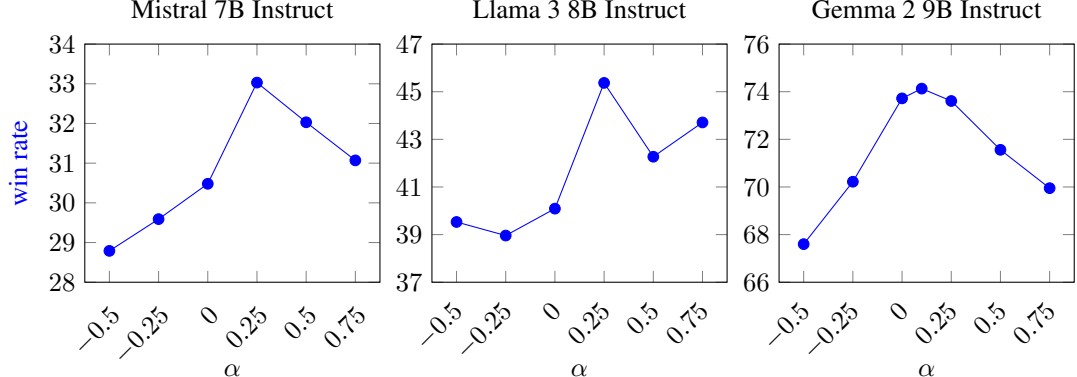

*Figure 8.* Effect of tuning $\alpha$ for AlphaPO. The highest AE2 length-controlled win rate is achieved for $\alpha > 0$: 0.1 for Gemma2 and 0.25 for the other two models.

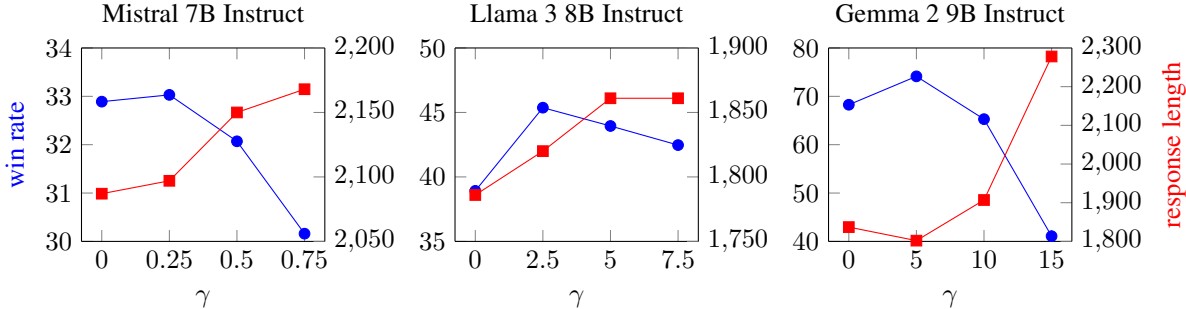

*Figure 9.* Effect of margin parameter $\gamma$ on AE2 length-controlled win rate and response length.

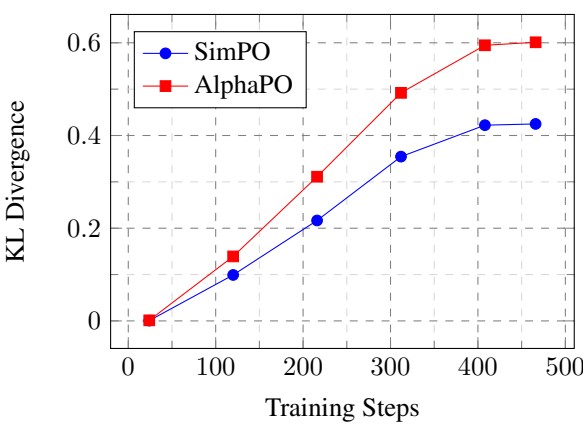

(a) KL Divergence evolution during training compared to original reference Mistral instruct model.

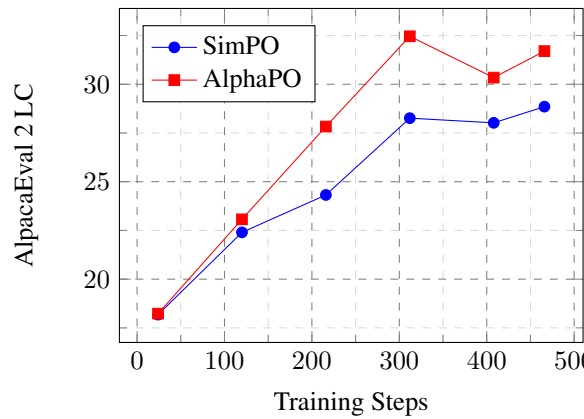

(b) AlpacaEval 2 length-controlled win rate evolution with training steps.

*Figure 10.* Side-by-side comparison of two metrics for Mistral 7B Instruct Model: KL Divergence (left) and AE (right).

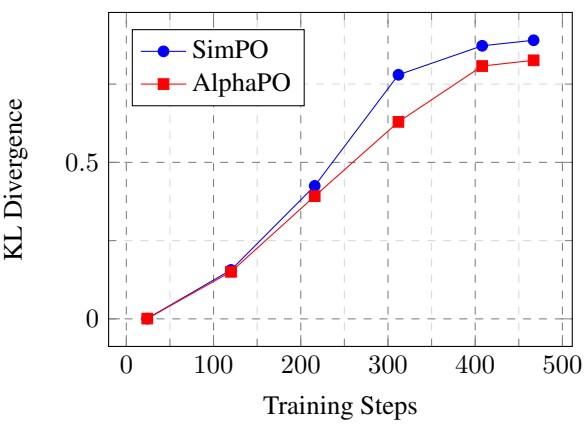

(a) KL Divergence evolution during training compared to original reference Mistral instruct model.

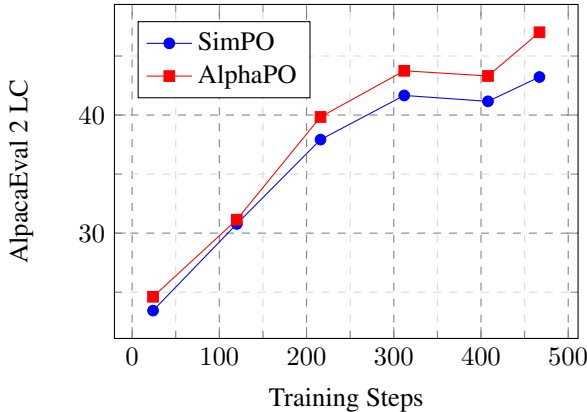

(b) AlpacaEval 2 length-controlled win rate evolution with training steps.

*Figure 11.* Side-by-side comparison of two metrics for Llama3 8B Instruct Model: KL Divergence (left) and AE (right).

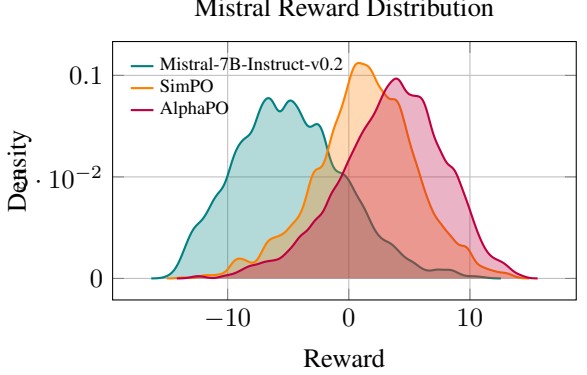

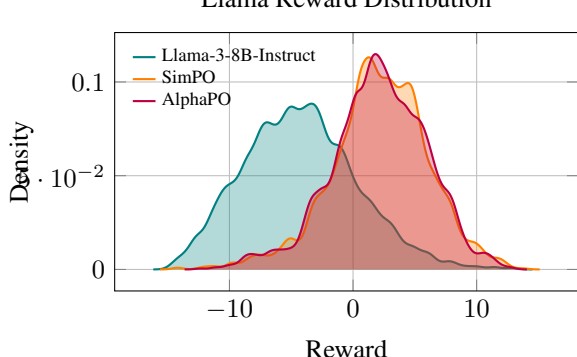

*Figure 12.* Comparison of reward distributions for PairRM for Mistral (left) and Llama 3 (right). AlphaPO yields a reward distribution that is either on par or better (i.e., right-shifted) when compared to strong baselines like SimPO.

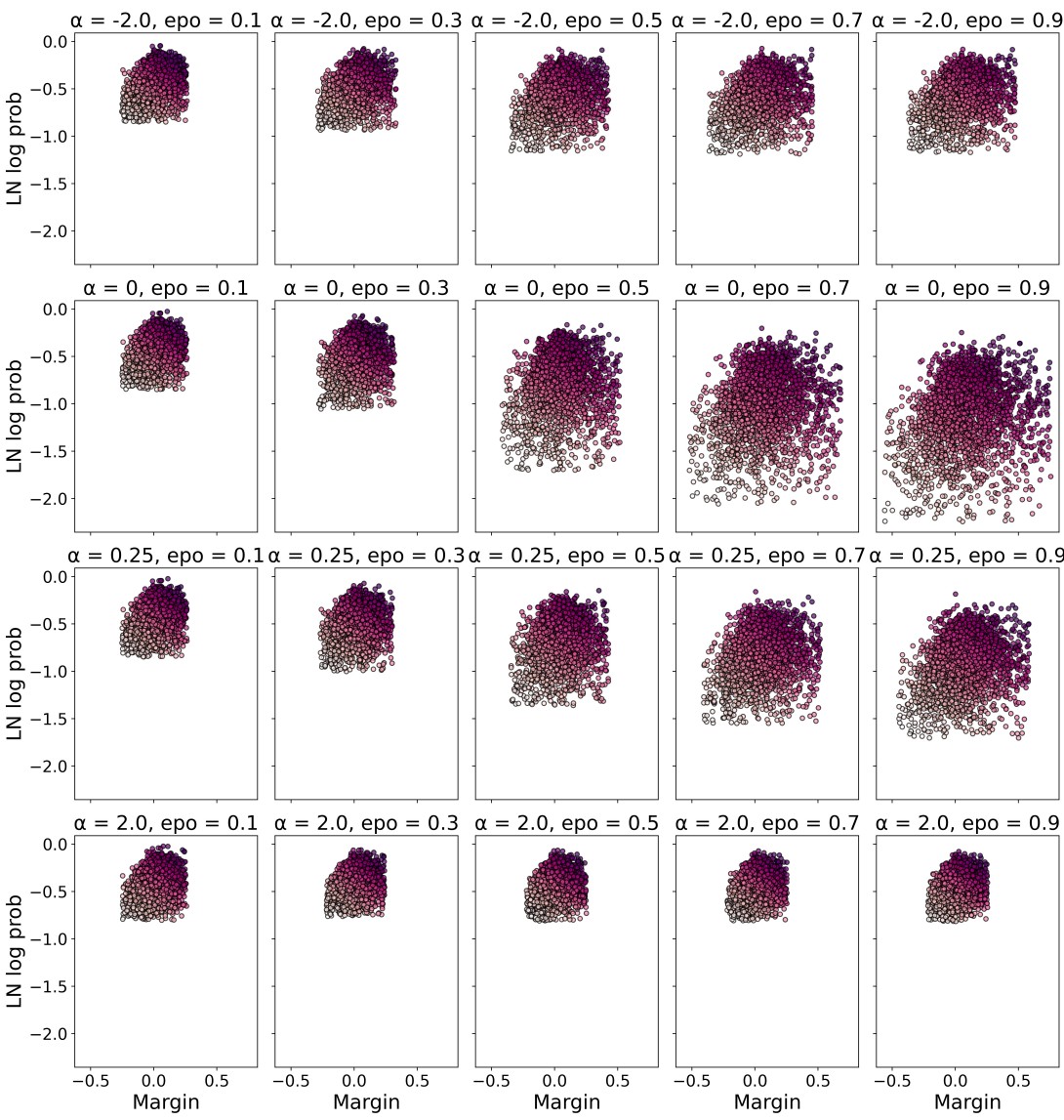

*Figure 13.* Scatter plot of length normalized preferred likelihood ($y$-axis) vs. length normalized margin ($x$-axis) for varying values of alpha for the **Mistral** instruct model. $\alpha$ is the alpha parameter, *epo* is epoch number. Outliers were moved before plotting. Clearly noticeable is the regularization effect of large absolute values of $\alpha$.

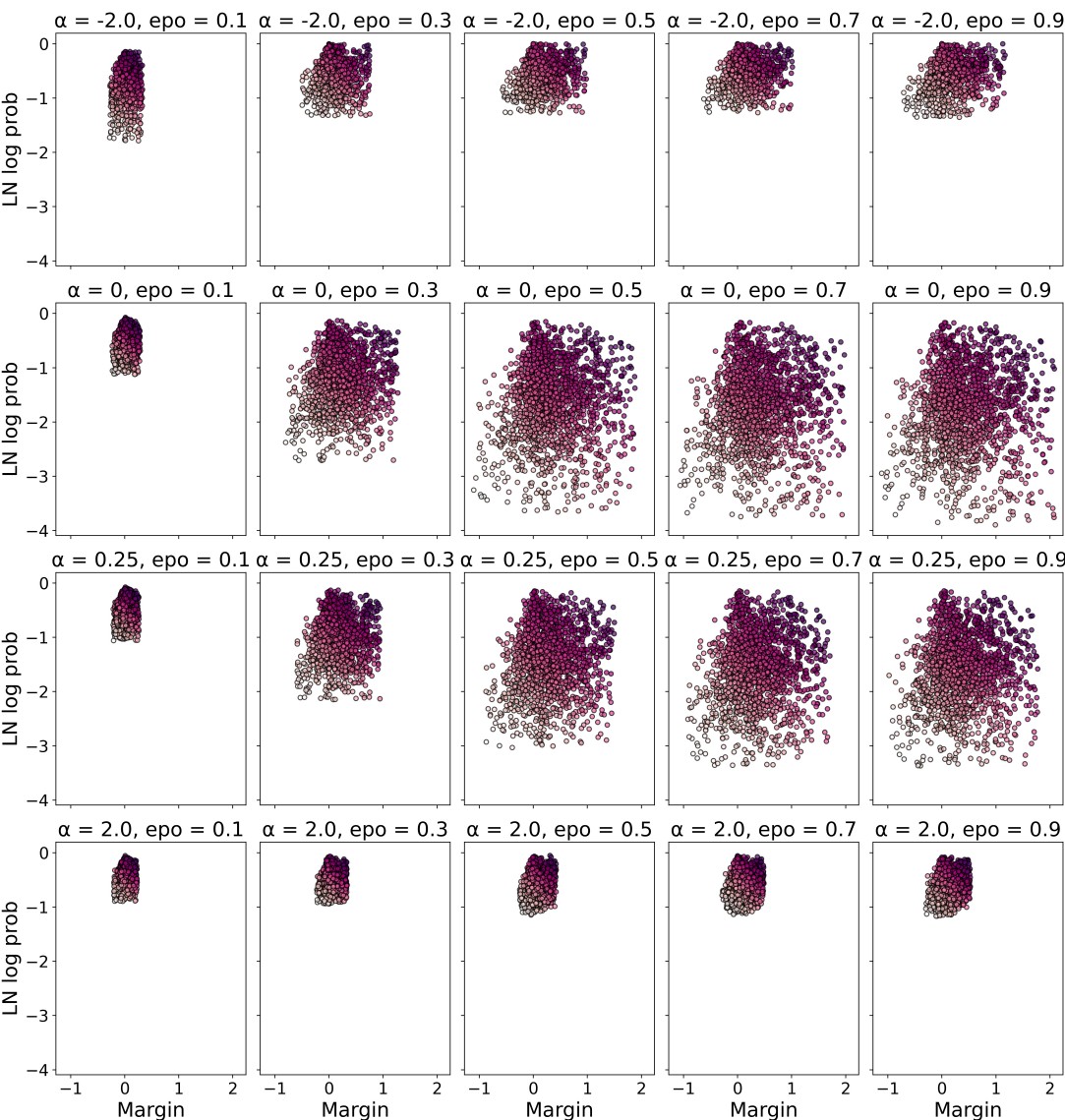

*Figure 14.* Scatter plot of length normalized preferred likelihood ($y$-axis) vs. length normalized margin ($x$-axis) for varying values of alpha for the **Gemma 2** instruct model. $\alpha$ is the alpha parameter, *epo* is epoch number. Outliers were moved before plotting. Clearly noticeable is the regularization effect of large absolute values of $\alpha$.

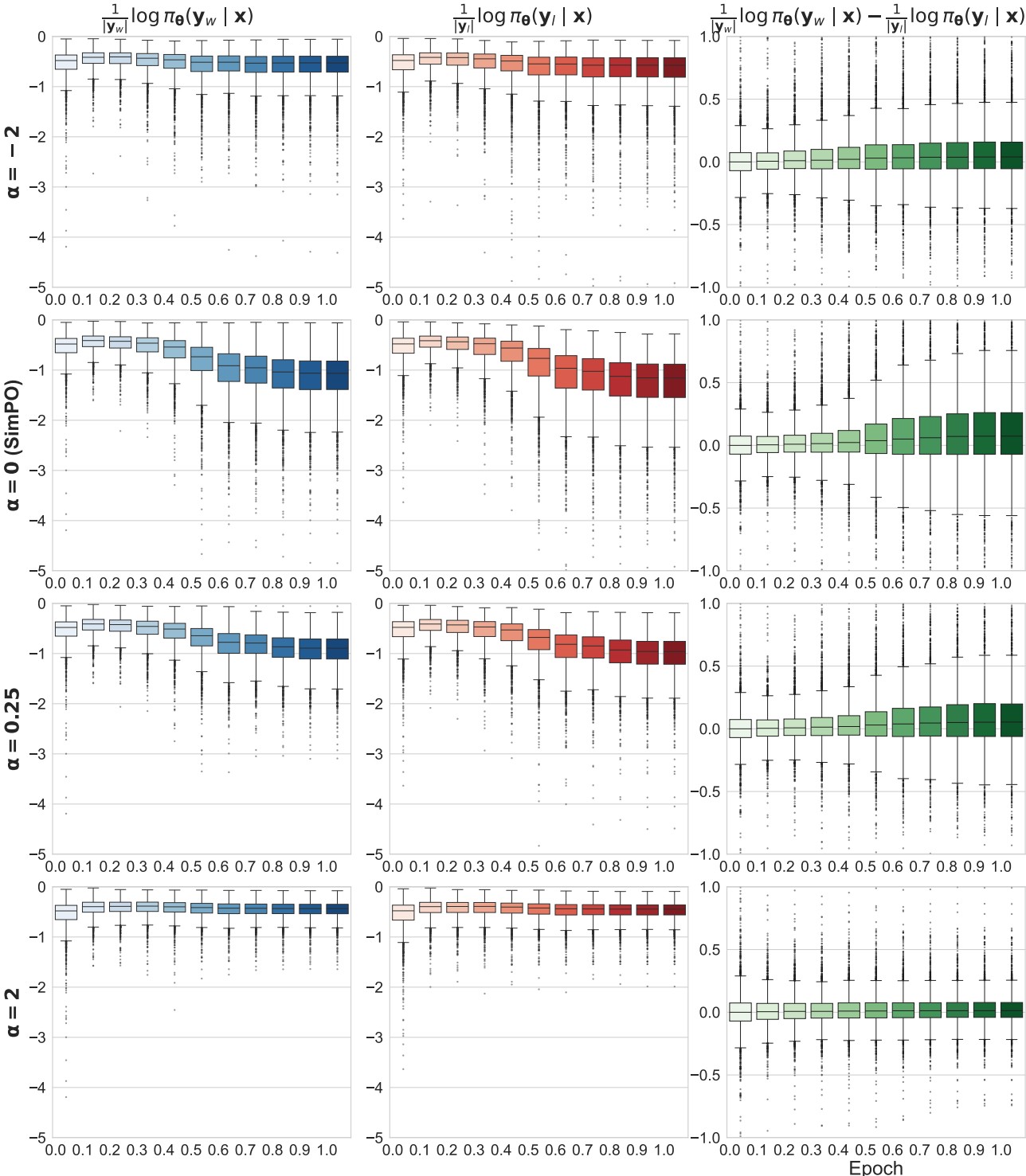

*Figure 15.* Mistral: Box plots illustrating the distribution of length-normalized preferred (dispreferred) likelihood and length-normalized margin across different $\alpha$ values. The plot provides a detailed view of variations in scatter intensity and performance metrics as a function of $\alpha$.

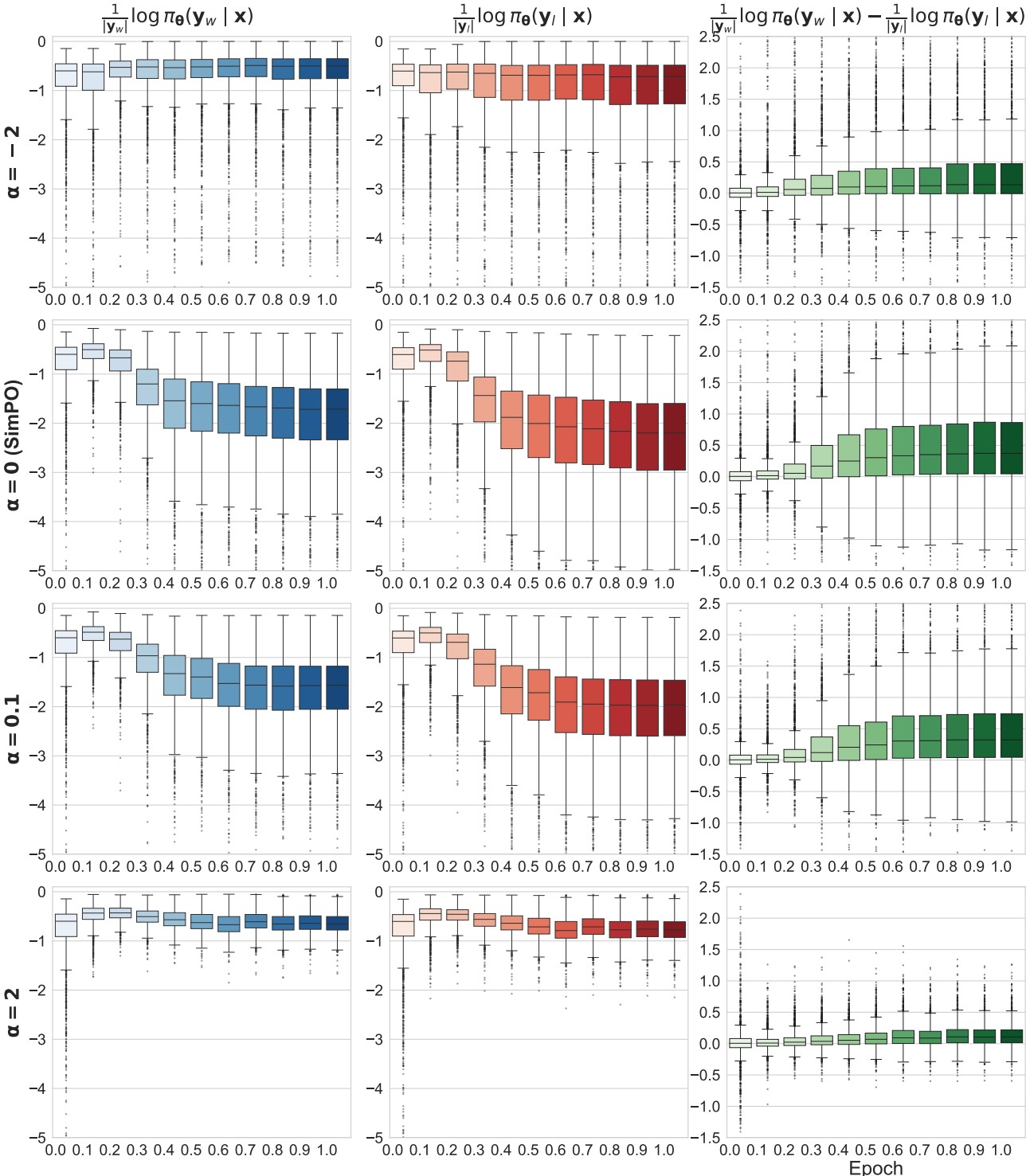

*Figure 16.* Gemma 2: Box plots illustrating the distribution of length-normalized preferred (dispreferred) likelihood and length-normalized margin across different $\alpha$ values. The plot provides a detailed view of variations in scatter intensity and performance metrics as a function of $\alpha$.

